# Structural basis for pegRNA-guided reverse transcription by a prime editor

Yutaro Shuto[1,7], Ryoya Nakagawa[1,7 ✉], Shiyou Zhu[2,3,4,5,6], Mizuki Hoki[1], Satoshi N. Omura[1], Hisato Hirano[1], Yuzuru Itoh[1], Feng Zhang[2,3,4,5,6] & Osamu Nureki[1 ✉]

The prime editor system composed of *Streptococcus pyogenes* Cas9 nickase (nSpCas9) and engineered Moloney murine leukaemia virus reverse transcriptase (M-MLV RT) collaborates with a prime editing guide RNA (pegRNA) to facilitate a wide variety of precise genome edits in living cells[1]. However, owing to a lack of structural information, the molecular mechanism of pegRNA-guided reverse transcription by the prime editor remains poorly understood. Here we present cryo-electron microscopy structures of the SpCas9–M-MLV RTΔRNaseH–pegRNA–target DNA complex in multiple states. The termination structure, along with our functional analysis, reveals that M-MLV RT extends reverse transcription beyond the expected site, resulting in scaffold-derived incorporations that cause undesired edits at the target loci. Furthermore, structural comparisons among the pre-initiation, initiation and elongation states show that M-MLV RT remains in a consistent position relative to SpCas9 during reverse transcription, whereas the pegRNA–synthesized DNA heteroduplex builds up along the surface of SpCas9. On the basis of our structural insights, we rationally engineered pegRNA variants and prime-editor variants in which M-MLV RT is fused within SpCas9. Collectively, our findings provide structural insights into the stepwise mechanism of prime editing, and will pave the way for the development of a versatile prime editing toolbox.

The CRISPR RNA-guided endonuclease Cas9 binds to a single guide RNA (sgRNA) and cleaves double-stranded DNA (dsDNA) targets complementary to the RNA guide[2]. Consequently, CRISPR–Cas9-based approaches have been harnessed for genome editing in eukaryotic cells[3]. A versatile genome-editing approach—prime editing—has been developed that allows virtually any desired base substitution, small insertion or small deletion to be installed into a genome at a specific site, without requiring double-stranded breaks or donor DNA templates[1]. Accordingly, prime editing can potentially correct the vast majority of known pathogenic mutations. Indeed, prime editing has been broadly applied to install precise mutations in human cells[4] and in various organisms, such as plants[5], zebrafish[6], mice[7] and *Drosophila*[8].

The prime editing system has two components: a prime editor composed of a *Streptococcus pyogenes* Cas9 nickase (nSpCas9) and an engineered Moloney murine leukaemia virus reverse transcriptase (M-MLV RT); and a prime editing guide RNA (pegRNA) with an sgRNA region and a 3′ extension region[1] (Fig. 1a). The sgRNA region consists of a guide sequence for targeting the specific site and a scaffold for interacting with SpCas9, whereas the 3′ extension region includes a reverse transcription template (RTT) followed by a primer-binding site (PBS). The PBS has 10–15-nucleotide (nt) sequences complementary to the 3′ end of a non-target strand (NTS), and the RTT encodes the desired edits (Fig. 1a). In prime editing, nSpCas9 recognizes dsDNA targets at a sequence complementary to the guide segment in the sgRNA and

flanked by an NGG (where N is any nucleotide) protospacer adjacent motif (PAM), and nicks the NTS. Then, M-MLV RT binds to the PBS–NTS heteroduplex and reverse transcribes the RTT sequence to incorporate the desired edits into the target loci (Fig. 1b).

However, the mechanism by which the prime editor recognizes the PBS–NTS heteroduplex to initiate and terminate reverse transcription of the RTT sequence remains poorly understood, owing mainly to the lack of structural information. To address this question, we determined cryo-electron microscopy (cryo-EM) structures of the prime editor in multiple states, providing a structural framework for understanding this innovative genome engineering system.

## Determining the structure of the prime editor

We purified endogenous prime editor 2 (PE2) (nSpCas9 fused with engineered M-MLV RT) and performed an in vitro prime editing assay using a pegRNA (28-nt RTT and 13-nt PBS) and 5′-Cy5-labelled pre-nicked dsDNA substrates (Fig. 1c, Extended Data Fig. 1a–c and Supplementary Table 1). PE2 generated DNA products by reverse transcription of the RTT sequence (Fig. 1c). To assemble a complex stalled at the termination of reverse transcription, we incubated PE2 with the target DNA and a pegRNA (5′-UCACAG-3′ RTT and 13-nt PBS) designed to halt reverse transcription at the 5′ end of the RTT sequence, using 2′,3′-deoxyadenosine 5′-triphosphate (ddATP) (Extended Data Fig. 1c and

[1]Department of Biological Sciences, Graduate School of Science, The University of Tokyo, Tokyo, Japan. [2]Broad Institute of MIT and Harvard, Cambridge, MA, USA. [3]McGovern Institute for Brain Research at MIT, Massachusetts Institute of Technology, Cambridge, MA, USA. [4]Department of Biological Engineering, Massachusetts Institute of Technology, Cambridge, MA, USA. [5]Department of Brain and Cognitive Science, Massachusetts Institute of Technology, Cambridge, MA, USA. [6]Howard Hughes Medical Institute, Massachusetts Institute of Technology, Cambridge, MA, USA. [7]These authors contributed equally: Yutaro Shuto, Ryoya Nakagawa. ✉e-mail: ryoya.nakagawa@bs.s.u-tokyo.ac.jp; nureki@bs.s.u-tokyo.ac.jp

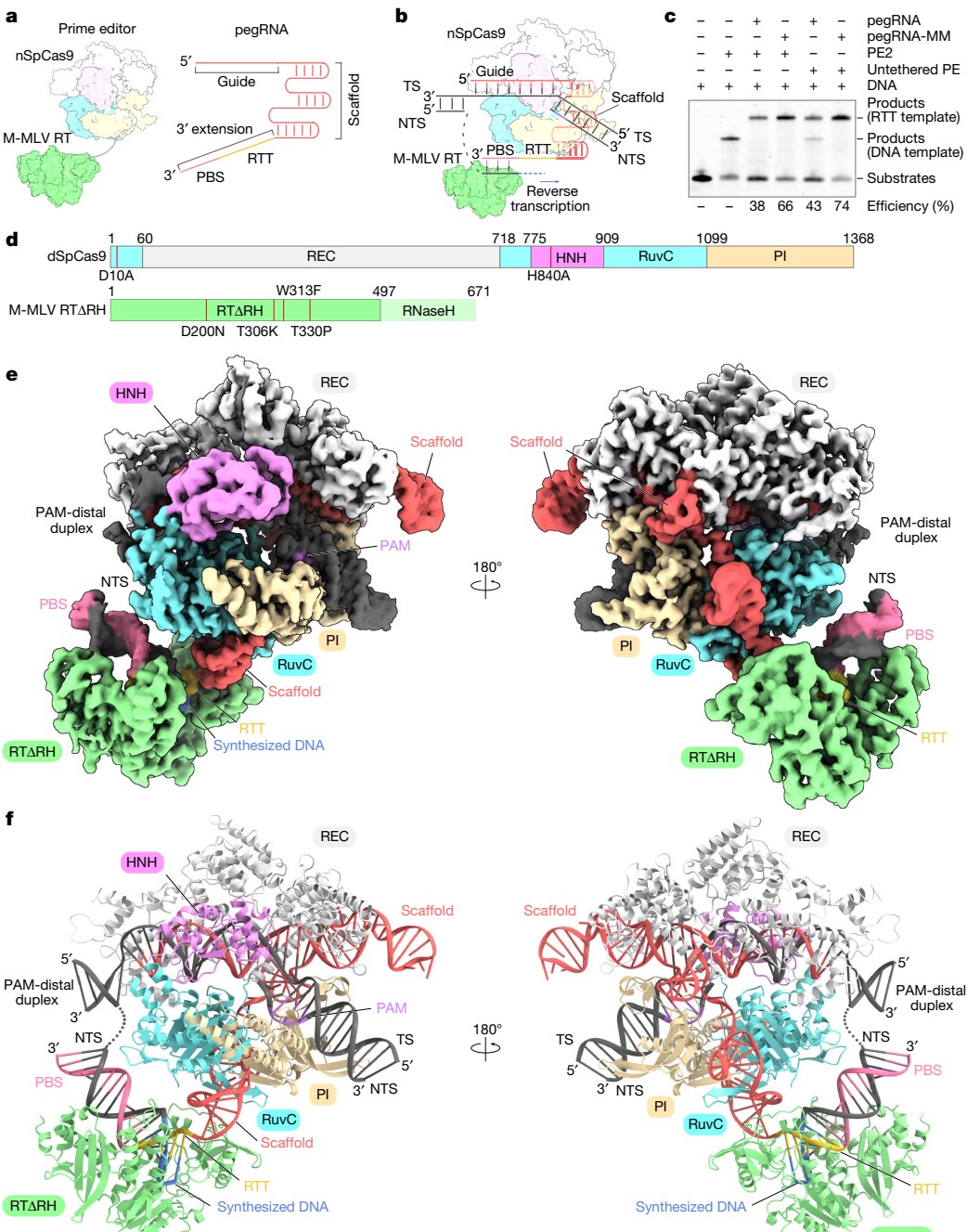

**Fig. 1 | Cryo-EM structure of the prime editor in the termination state.**
**a**, Two components of the prime editing system. The prime editor is composed of nSpCas9 and M-MLV RT, and the pegRNA comprises the sgRNA region and the 3′ extension region. The 3′ extension region consists of the PBS and the RTT. The sgRNA region, PBS and RTT are coloured red, pink, and yellow respectively. **b**, Schematic of the nSpCas9–M-MLV RT–pegRNA–target DNA complex. nSpCas9 nicks the NTS in guide RNA- and PAM-dependent manners, and then M-MLV RT binds to the PBS–NTS heteroduplex and reverse transcribes the RTT sequences. TS, target strand. **c**, In vitro prime editing assay using purified PE2, pegRNA containing 13-nt PBS and 28-nt RTT, and 5′-Cy5-labelled pre-nicked DNA substrates. The PE2–pegRNA complex was mixed with DNA substrates, and incubated at 37 °C for 10 min. The reaction products were separated on a 10% Novex PAGE TBE–urea gel, and the Cy5 fluorescence was then visualized. PegRNA-MM refers to a pegRNA designed with non-complementary sequences between the guide and the PBS. Untethered PE refers to a construct in which nSpCas9 and M-MLV RTΔRNaseH (RTΔRH) were purified separately. Untethered PE exhibits pegRNA-dependent reverse transcription activity comparable to that of PE2. The experiments were repeated three times with similar results. **d**, Domain structures of dSpCas9 (D10A/H840A) and RTΔRH (D200N/T306K/W313F/T330P). RTΔRH lacks the RNaseH domain (residues 498–671). PI, PAM-interacting domain. **e**, Cryo-EM density map of the SpCas9–RTΔRH–pegRNA–target DNA complex in the termination state. **f**, Overall structure of the SpCas9–RTΔRH–pegRNA–target DNA complex in the termination state. The disordered regions are indicated as dotted lines.

Supplementary Table 2). However, we were unable to obtain a high-resolution cryo-EM density map, owing probably to sample heterogeneity. To overcome this issue, we made two modifications. First, we used a modified pegRNA designed with a non-complementary sequence between the PBS and the guide, resulting in increased target products and decreased by-products[9,10] (pegRNA-MM; Fig. 1c and Extended Data Fig. 1c). Second, we separately purified a catalytically inactive SpCas9 (D10A and H840A) and an engineered M-MLV RTΔRNaseH

(referred to as RTΔRH for simplicity), instead of PE2 (Fig. 1d and Extended Data Fig. 1a). Our in vitro prime editing assay showed that this untethered prime editor (untethered PE) exhibits a comparable prime editing efficiency to that of PE2, consistent with previous studies in mammalian cells[9,11] (Fig. 1c). With these modifications, we reconstituted the termination state of the SpCas9–RTΔRH–pegRNA–target DNA complex using ddATP, and successfully obtained a three-dimensional (3D) reconstruction with an overall resolution of 3.0 Å (Fig. 1d–f and Extended Data Fig. 2a–e). To improve the local resolution of the RTΔRH, we then performed local refinement focusing on the RTΔRH-proximal region, which yielded a 3.5-Å local map (Extended Data Fig. 2f,g). Finally, we combined these maps to generate a full model of the SpCas9–RTΔRH–pegRNA–target DNA complex (Fig. 1e,f, Extended Data Fig. 2c,h and Extended Data Table 1).

## Overall structure of the prime editor

The cryo-EM structure reveals that SpCas9 assembles with a scaffold region of the pegRNA (G21–C96) to form a ribonucleoprotein, and binds to the target DNA in guide RNA-dependent and PAM-dependent manners, as previously observed in the SpCas9–sgRNA–target DNA structure[12] (Protein Data Bank (PDB): 7Z4L, root-mean-square deviation (RMSD) = 1.38 Å for 1,298 equivalent Cα atoms) (Fig. 1f and Extended Data Fig. 3a). This observation suggests that the 3′ extension region of the pegRNA and M-MLV RT do not inhibit target DNA recognition by SpCas9, consistent with a previous study[1]. The 3′ extension region (U97–G115) is resolved in our density map, and forms an RNA–DNA heteroduplex with the NTS on a weakly positively charged surface facing the RuvC domain of SpCas9 (Fig. 1e,f and Extended Data Fig. 3b,c). Nucleotides C103–G115 in the 13-nt PBS base-pair with nucleotides dC(−16*)–dG(−4*) in the NTS to form a PBS–NTS heteroduplex (Fig. 2a,b). The clear density revealed that nucleotides C98–G102 in the 6-nt RTT base-pair with the newly synthesized (reverse-transcribed) nucleotides dC1†–dG5† in the NTS, and that U97, located at the 5′ end of the RTT, is kinked from the scaffold region and forms a base pair with ddA6† in the NTS (Fig. 2a–c). The RTΔRH is clearly visible in the density map, except for the peripheral regions (residues 1–23, 449–454 and 483–496) (Extended Data Fig. 3d). The RTΔRH binds to the PBS–NTS and RTT–synthesized DNA heteroduplexes through the positively charged central groove (Extended Data Fig. 3e). Notably, the M-MLV RT catalytic motif (YVDD) is located close to the last U97-ddA6† base pair, indicating that this structure captures the state in which the prime editor has just completed reverse transcription up to the end of the RTT sequence (Fig. 2c and Extended Data Fig. 3e).

## RT proceeds beyond RTT

Given that reverse transcription of the 3′ extension of the pegRNA by M-MLV RT can proceed into the scaffold region, the precise termination site of reverse transcription and its termination mechanism in prime editing remain unknown. In our structure, although the RTΔRH proceeds with reverse transcription up to the end of the RTT, there is an approximate 10-Å separation between SpCas9 and RTΔRH, leaving sufficient space for further reverse transcription (Fig. 3a). In addition, we observed that C96, at the 3′ end of the scaffold region, forms extensive interactions with key residues of RTΔRH that are crucial for the processivity of reverse transcription[13,14] (Fig. 3b). In particular, the ribose moiety of C96 stacks and hydrogen bonds with L99 and R116, respectively, whereas the G82-C96 base pair forms a stacking interaction with Y64 (Fig. 3b). These structural observations suggest that, if not halted by ddATP, the RTΔRH would continue reverse transcription beyond the RTT into the scaffold region. To biochemically characterize the termination site of reverse transcription, we performed in vitro prime editing assays using the PE2 and the untethered PE. With both constructs, we observed that the reverse transcription products with dNTPs were consistently three nucleotides longer than those with ddATP, regardless

of the length of the RTT sequence (Fig. 3c and Extended Data Fig. 4a). These results indicate that reverse transcription by the PE2 and the untethered PE does not terminate at the RTT terminus, but instead progresses up to U94 of the scaffold region. Our structure shows that U94 of the pegRNA is positioned close to SpCas9, suggesting that M-MLV RT may not proceed further (Fig. 3d). These biochemical and structural observations show that M-MLV RT extends reverse transcription up to U94 of the pegRNA, three nucleotides upstream of the RTT, and terminates reverse transcription by dissociating from the pegRNA, owing probably to steric hindrance with SpCas9.

Consistent with our finding that reverse transcription proceeds beyond the 3′ extension, previous studies reported that the prime editor induces scaffold-derived short (1- to 3-nt) incorporations that cause undesired edits at the target loci[1,15,16]. To eliminate these incorporations, we sought to engineer a pegRNA variant with modified scaffold sequences that are excessively reverse transcribed. In the structure, nucleotides U94–C96 base-pair with nucleotides A84–G82 to form a stem loop structure and are recognized by SpCas9 through non-base-specific interactions (Fig. 2a and Extended Data Fig. 4b). We thus hypothesized that pegRNA variants, designed by modifying U94–C96 to match the target locus and adjusting A84–G82 to maintain the stem structure, could efficiently trigger pegRNA-dependent reverse transcription while eliminating scaffold-derived short incorporations (Extended Data Fig. 4c). We first performed an in vitro prime editing assay using the wild-type pegRNA and three pegRNA variants with modified stem loop sequences, and confirmed that these modifications do not affect the pegRNA-dependent reverse transcription activity (Extended Data Fig. 4d). We next evaluated the prime editing efficiencies using PE2 and prime editor 3 (PE3) systems at five and four previously validated target conditions in HEK293 cells, respectively[1]. Consistent with our in vitro assay, modified pegRNAs successfully induced the desired edits at frequencies comparable to those of the wild-type pegRNA (Extended Data Fig. 4e). However, the modified pegRNAs also induced undesired incorporations at levels comparable to those of the wild-type pegRNA, with these insertions derived from the RTT sequence or scaffold sequences longer than three nucleotides (Extended Data Fig. 4e). These results suggest that these types of incorporations are dominant in the target sites, and that this strategy would only be effective at target sites where the short scaffold-derived incorporations are prevalent.

## RT primes at a specific position

Next, to investigate how the prime editor initiates reverse transcription, we designed a pegRNA (5′-GCACAU-3′ RTT and 13-nt PBS) that allows the incorporation of the first nucleotide ddA (Extended Data Fig. 1c and Supplementary Table 2). We reconstituted the initiation complex of the SpCas9–RTΔRH–pegRNA–target DNA using ddATP, and successfully obtained a 3D reconstruction with an overall resolution of 3.1 Å (Extended Data Fig. 5a–e). Unexpectedly, we observed density corresponding to the RTΔRH in this map, although the density was relatively poor (Extended Data Fig. 5d). This observation suggests that M-MLV RT is located at a fixed position relative to SpCas9 at the initiation of reverse transcription. We performed local refinement focusing on the RTΔRH-proximal region, and finally obtained a composition map of the initiation complex (Fig. 4a,b, Extended Data Fig. 5f–h and Extended Data Table 1). In the final map, we identified the characteristic Cα atoms of the RTΔRH, and modelled its structure into the density map as a single rigid body. The PBS (C103–G115) forms the RNA–DNA heteroduplex with the NTS (dC[−16*]–dG[−4*]), and is sandwiched between the RuvC domain of SpCas9 and RTΔRH (Fig. 4b and Extended Data Fig. 6a–c). U102 at the 3′ end of the RTT forms a base pair with ddA1†, whereas the rest (C98–C100) is disordered except for G97 and A101, suggesting the flexibility of the RTT before reverse transcription (Fig. 4b and Extended Data Fig. 6a–c). In the structure, although RTΔRH is located

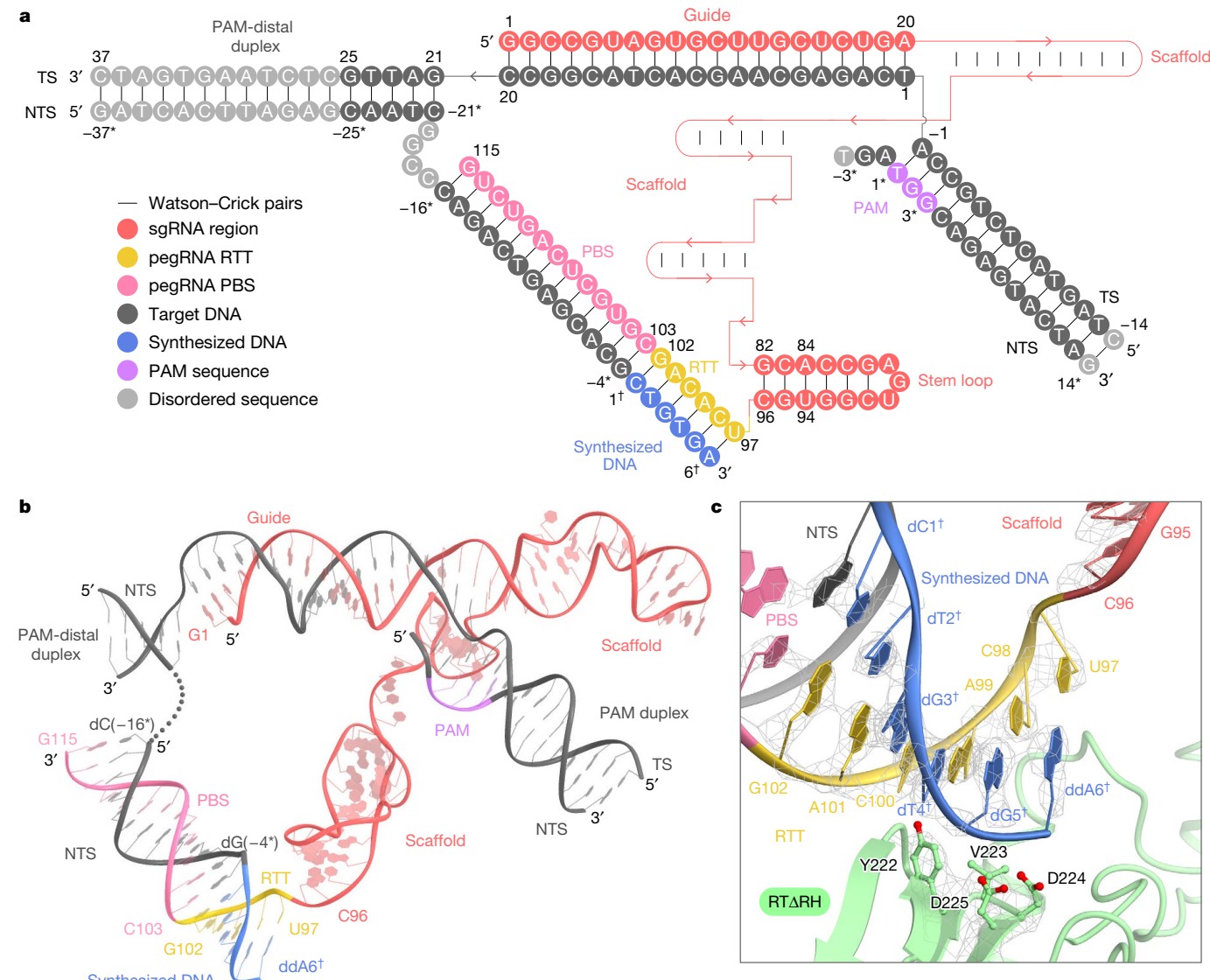

**Fig. 2 | Nucleic acid architecture. a**, Schematic of the pegRNA and target DNA in the termination state. Except for the 3′ stem loop (G82–C96), the scaffold region of the pegRNA is represented with a red line for simplicity. The disordered regions are coloured grey. **b**, Structure of the pegRNA and target DNA in the termination state. The disordered nucleotides G(−20*)–C(−17*) of the NTS are indicated with a dotted line. **c**, Close-up view of the M-MLV RT active site. The cryo-EM densities for the catalytic residues (YVDD), the RTT–newly synthesized DNA heteroduplex (U97-ddA6†–G102-dC1†) and the 3′ end of the stem loop (G95 and C96) are shown as grey meshes.

at a fixed position relative to SpCas9, it lacks direct interactions with SpCas9 and the scaffold region of the pegRNA. Instead, RTΔRH forms extensive interactions with the PBS–NTS heteroduplex, with the active site positioned near the U102-ddA1† base pair (Fig. 4a,b and Extended Data Fig. 6b,c). These structural observations suggest that the position of the PBS–NTS heteroduplex might have a crucial role in defining the initiation point for reverse transcription. To validate this hypothesis, we reconstituted the SpCas9–pegRNA–target DNA complex without RTΔRH and attempted to determine the cryo-EM structure of its pre-initiation complex (Supplementary Table 2). Considering that the length of the RTT sequence might restrict the position of the PBS–NTS heteroduplex, we used a pegRNA with a long RTT sequence (28 nt) for structural determination (Extended Data Fig. 1c). We obtained a 3D reconstruction of the pre-initiation complex with an overall resolution of 3.2 Å (Fig. 4c,d and Extended Data Fig. 7a–e). In the density map, we observed a rod-shaped density corresponding to the PBS–NTS heteroduplex on the interface facing the RuvC domain (Fig. 4c,d and Extended Data Fig. 7f), suggesting that the PBS–NTS heteroduplex

stably resides on this surface. This is possibly because the helical duplex conformation imposes topological constraints that align the positively charged surface of the RuvC domain, favouring this arrangement. A structural comparison of the pre-initiation and initiation complexes revealed that the positions of the PBS–NTS heteroduplex are similar in both states (Fig. 4a–d and Extended Data Fig. 7f–h). These structural observations indicate that the PBS–NTS heteroduplex is formed to face the RuvC domain, and that M-MLV RT then recognizes and binds to the heteroduplex to initiate reverse transcription of the RTT sequence.

## RT proceeds keeping its position

Finally, to capture an elongation state of the prime editor, we prepared a pegRNA with a 28-nt RTT, designed to halt reverse transcription at the 16th nucleotide with ddATP, and analysed its structure by cryo-EM (Extended Data Fig. 1c). Using 3D classification and local refinement, we obtained a composite density map of the elongation complex (16-nt) from an overall map at 3.1 Å resolution and a local map around

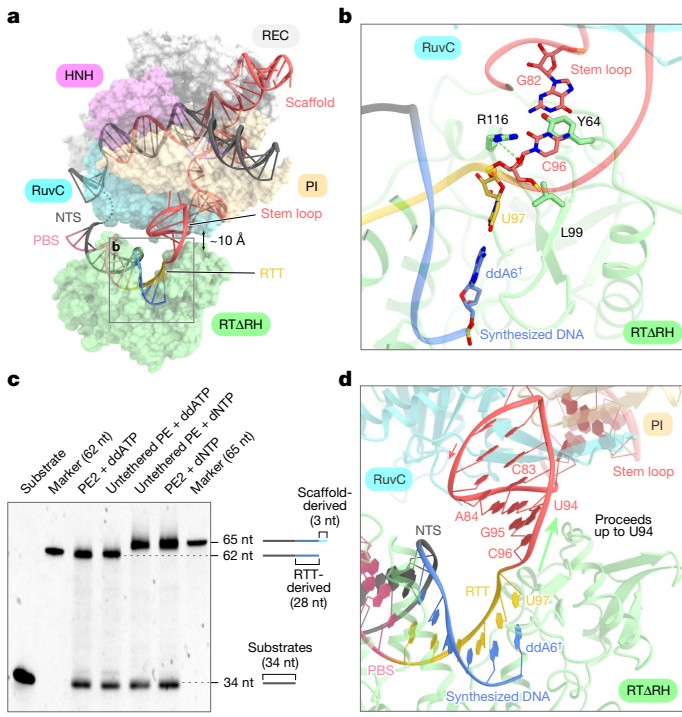

**Fig. 3 | Termination site of reverse transcription. a**, Surface representation of the SpCas9–RTΔRH–pegRNA–target DNA complex in the termination state. Although reverse transcription proceeds up to the end of the RTT, there is an approximate 10-Å separation between SpCas9 and RTΔRH, leaving sufficient space for further reverse transcription. **b**, Recognition of the 5′ end of the RTT sequence and 3′ end of the stem loop. The key residues Y64/L99 and R116, crucial for the processivity of reverse transcription, form van der Waals interactions and a hydrogen bond with C96, respectively. The hydrogen bond is depicted with a green dashed line. **c**, In vitro prime editing assay using PE2 or untethered PE, a pegRNA and 5′-Cy5-labelled pre-nicked DNA substrates. The RTT sequence of the pegRNA was designed to contain 'U' only at the 5′ end, enabling reverse transcription to stop at the end of the RTT sequence when using ddATP. The PE2/untethered PE–pegRNA complex was added to DNA substrates with dNTPs or with dCTP, dTTP, dGTP and ddATP (referred to as ddATP in **c** for simplicity), and incubated at 37 °C for 10 min. The reaction products were separated on a 15% Novex PAGE TBE–urea gel, and the Cy5 fluorescence was visualized. The experiments were repeated three times with similar results. **d**, Close-up view of the space sandwiched between SpCas9 and M-MLV RT. SpCas9 recognizes the stem loop region (G82–C96) through non-base-specific interactions, and M-MLV RT terminates reverse transcription at U97. There is sufficient space between SpCas9 and M-MLV RT for M-MLV RT to proceed up to U94.

RTΔRH at 6.1 Å resolution (Fig. 4e,f, Extended Data Fig. 8a–f, Supplementary Table 2 and Extended Data Table 1). As expected, we observed the density corresponding to the long RNA–DNA heteroduplex on the surface of the RuvC domain, with RTΔRH bound at its end (Fig. 4e,f). A structural comparison between the initiation and elongation complexes revealed that, despite reverse transcription progressing by 15 nt, the arrangement of the RTΔRH relative to the SpCas9 remains largely unchanged (Fig. 4b,f and Extended Data Fig. 8g). This observation suggests that, during the reverse transcription of the RTT, M-MLV RT maintains a consistent position around the initiation site. Thus, we hypothesized that PE2 variants, in which M-MLV RT is tethered within SpCas9 by a short linker to anchor it close to the initiation site, would sufficiently induce pegRNA-dependent reverse transcription. To test this hypothesis, we designed two PE2 variants in which RTΔRH was inserted between G1054 and E1055 or T1068 and G1069 in the RuvC domain, connected by five amino-acid linkers (Fig. 4g and Extended Data Fig. 8h). We successfully purified these two variants and found that

they exhibit comparable reverse transcription efficiencies to that of the wild-type PE2 (Fig. 4h and Extended Data Fig. 1a). This result supports our hypothesis that reverse transcription of the RTT sequence consistently occurs around the surface of the RuvC domain. By contrast, the structural comparison also revealed that, owing to the formation of the RTT–synthesized DNA heteroduplex, the PBS–NTS heteroduplex is pushed in the opposite direction to the RTΔRH, resulting in the rearrangement of the PAM-distal target DNA duplex (Fig. 4i). We observed further movement of the PBS–NTS heteroduplex and additional rearrangement of the PAM-distal target DNA duplex in a state where reverse transcription had progressed up to the 28-nt RTT terminus (Fig. 4i, Extended Data Figs. 1c and 9a–h, Supplementary Table 2 and Extended Data Table 1). These structural and biochemical observations indicate that M-MLV RT consistently performs the reverse transcription of the RTT sequence at the initiation site, and that the RTT–synthesized DNA heteroduplex builds up along the longitudinal surface of SpCas9, which leads to the rearrangement of the PAM-distal target DNA duplex.

## Structure of M-MLV RT with substrate

Although M-MLV RT is widely used commercially, it has not been fully characterized owing to a lack of structural information about the substrate-binding state[13]. Our termination structure provides high-resolution insights into the substrate recognition of M-MLV RT. M-MLV RT comprises a polymerase region with palm, finger and thumb domains, along with connection and RNaseH domains, although our construct lacks the RNaseH domain (Extended Data Fig. 10a,b). A structural comparison between the substrate-bound and unbound states (PDB: 4MH8; ref. 17) revealed structural rearrangements in the finger, thumb and connection domains after substrate binding, resulting in the formation of the binding groove for the substrate[17] (Extended Data Fig. 10c–e). Notably, the connection domain undergoes local conformational changes and recognizes the terminal regions of the RNA–DNA heteroduplex (Extended Data Fig. 10e). This observation indicates that the connection domain not only serves to connect the polymerase region and the RNaseH domain but also has a crucial role in substrate recognition, consistent with a previous functional analysis[18]. Although numerous techniques have been applied to enhance the efficiency of M-MLV RT (refs. 19–22), this structural information is expected to facilitate further rational modifications.

## Discussion

In this study, we determined the cryo-EM structures of the pre-initiation, initiation, elongation and termination states of the prime editor. These structural observations, together with our functional analyses, provide profound insights into the stepwise model for prime editing (Fig. 5). First, nSpCas9 recognizes the target DNA in guide RNA-dependent and PAM-dependent manners to nick the NTS, using a RuvC nuclease domain. Second, the PBS of the pegRNA base-pairs with the nicked NTS to form the PBS–NTS heteroduplex on the marginal surface of the RuvC domain, owing probably to topological constraints and electrostatic attraction (the pre-initiation state). Third, M-MLV RT recognizes the PBS–NTS heteroduplex on the surface, initiating the reverse transcription of the RTT sequence (the initiation state). Fourth, M-MLV RT consistently engages in reverse transcription of the RTT sequence around the initiation site, and the RTT–synthesized DNA heteroduplex accumulates along the longitudinal surface of SpCas9, accompanied by the rearrangement of the PAM-distal duplex (the elongation state). Fifth, when M-MLV RT has completed reverse transcription up to the end of the RTT sequence, some space remains between nSpCas9 and M-MLV RT to accommodate further reverse transcription of the scaffold sequence (the termination state). Then, M-MLV RT invades the scaffold region of the pegRNA, extending reverse transcription up to three nucleotides upstream

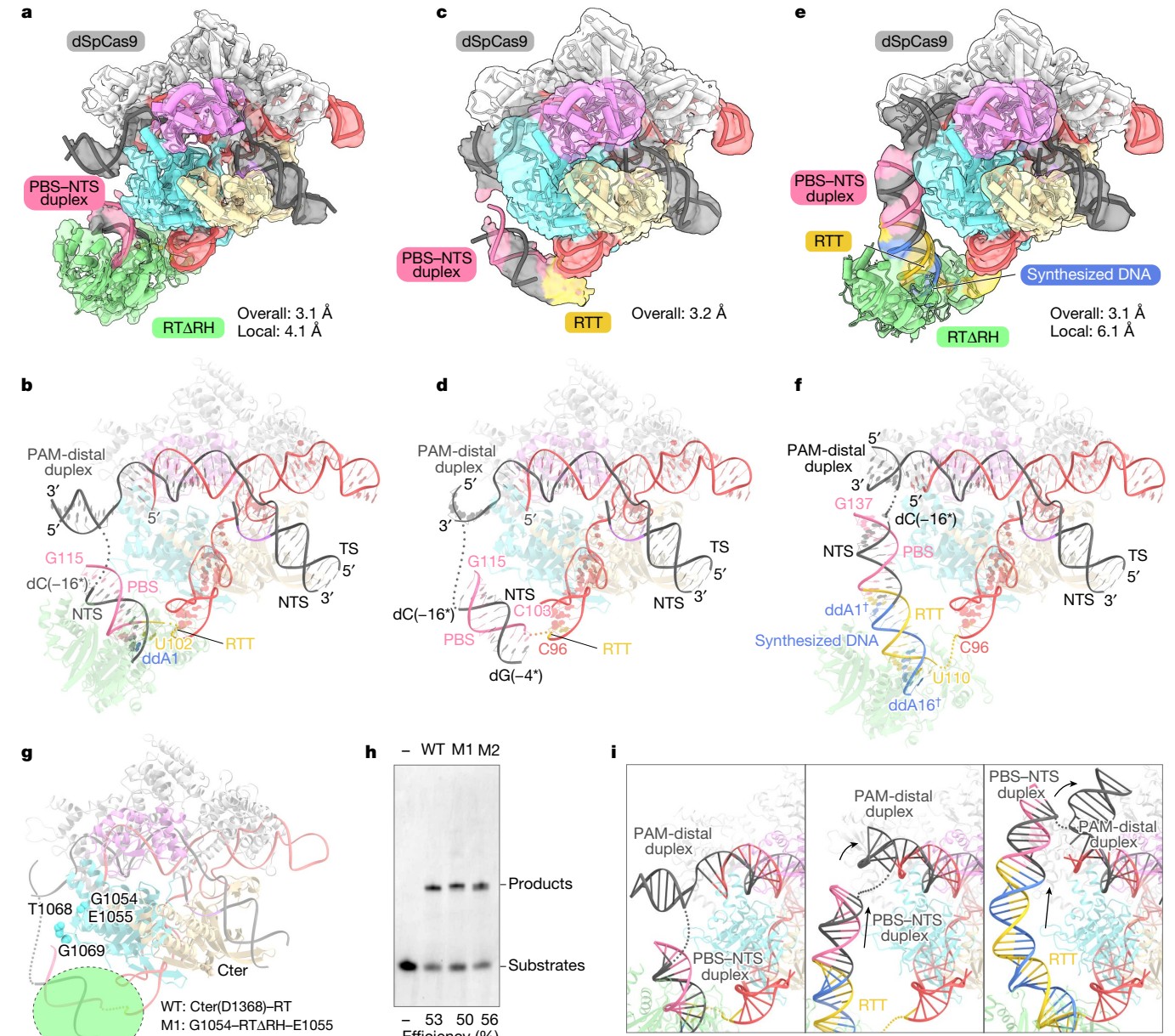

**Fig. 4 | Cryo-EM structures of the prime editor in multiple states. a–f**, Cryo-EM densities (**a**,**c**,**e**) and overall structures (**b**,**d**,**f**) of the SpCas9–RTΔRH–pegRNA–target DNA complex in the initiation state (**a**,**b**), the SpCas9–pegRNA–target DNA complex in the pre-initiation state (**c**,**d**) and the SpCas9–RTΔRH–pegRNA–target DNA complex in the elongation state (16-nt) (**e**,**f**). The disordered regions are indicated as dotted lines. **g**, Mapping of the G1054, E1055, T1068 and G1069 residues in the RuvC domain onto the SpCas9–pegRNA–target DNA complex in the pre-initiation state. These residues are located close to the PBS–NTS heteroduplex. **h**, In vitro prime editing assay using wild-type PE2 (referred to as WT) and two prime editor variants, with the RTΔRH inserted between G1054 and E1055 (M1: G1054–RTΔRH–E1055) or T1068 and G1069 (M2: T1068–RTΔRH–G1069) in the RuvC domain. The experiments were repeated three times with similar results. **i**, Close-up views of the PBS–NTS heteroduplex and the PAM-distal duplex in the initiation (left), elongation (16-nt) (middle) and elongation (28-nt) (right) states. As the reverse transcription of the RTT sequence progresses, the PBS–NTS heteroduplex is pushed in the opposite direction to the RTΔRH (shown with a black arrow), resulting in the rearrangement of the PAM-distal duplex.

of the RTT (U94), and dissociates from the pegRNA, owing to steric hindrance with nSpCas9 (this is speculative). Our in vitro prime editing assay revealed that the recently reported prime editor 6 (PE6) a–d also generate reverse transcription products with the same length as PE2, suggesting that this termination mechanism may be common among PE2 and PE6a–d (ref. 16) (Extended Data Fig. 10f). Sixth, the newly synthesized DNA containing the desired edit is incorporated into the genome, resulting in its permanent installation.

Our structural and in vitro prime editing analysis revealed that the prime editor does not terminate reverse transcription at the end of the RTT, but extends into the scaffold region. However, scaffold-derived incorporations in mammalian cells are much less frequent as compared with desired edits. These results imply that the reverse-transcribed DNA flaps are processed by various enzymes, such as endogenous exonucleases, before being inserted into the target site. A previous study reported that DNA mismatch repair inhibits the efficiency and precision of prime editing[15]. Further efforts focused on how reverse-transcribed DNA is properly incorporated into the target site will be important to fully understand the prime editing system and improve its efficiency. In addition, the prime editor induces undesired insertions derived from the RTT sequence or scaffold sequences longer than three nucleotides[6]. Elucidating the mechanisms of these insertions and developing

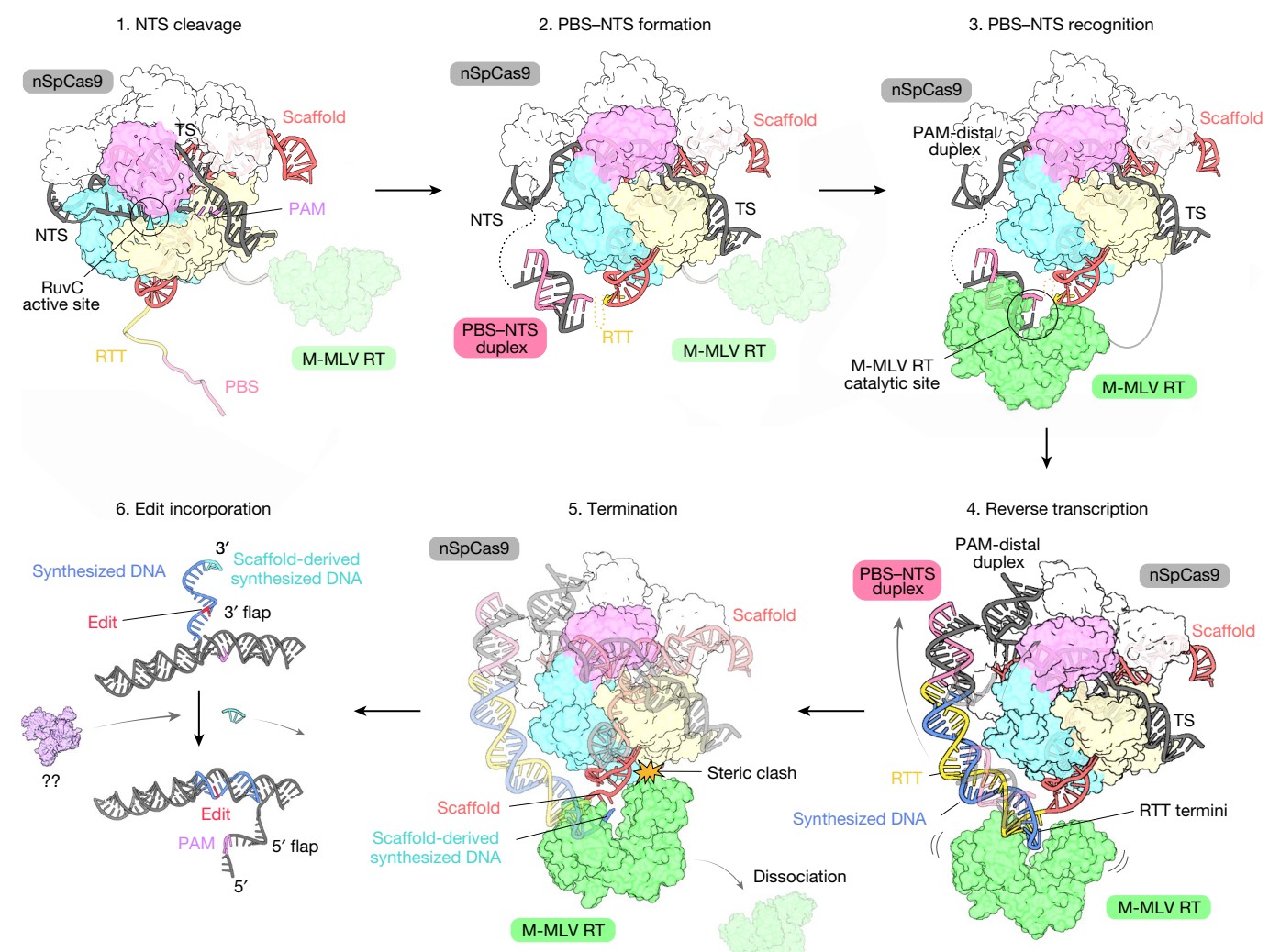

**Fig. 5 | Stepwise model of prime editing.** Structure-based stepwise model of prime editing. (1) NTS cleavage. nSpCas9 nicks the NTS in guide RNA-dependent and PAM-dependent manners. (2) PBS–NTS formation. The PBS of the pegRNA base-pairs with the nicked NTS to form the PBS–NTS heteroduplex on the marginal surface of the RuvC domain. (3) PBS–NTS recognition. M-MLV RT recognizes the PBS–NTS heteroduplex on the surface, initiating the reverse transcription of the RTT sequence. (4) Reverse transcription. M-MLV RT consistently engages in reverse transcription of the RTT sequence around the initiation site, and the RTT–synthesized DNA heteroduplex accumulates along the longitudinal surface of SpCas9, accompanied by the rearrangement of the PAM-distal duplex. (5) Termination. M-MLV RT does not terminate at the end of the RTT sequence, but instead invades the scaffold region of the pegRNA, extending the reverse transcription up to three nucleotides upstream of the RTT (U94). It is speculated that M-MLV RT dissociates from the pegRNA owing to steric hindrance with nSpCas9. (6) Edit incorporation. The newly synthesized DNA containing the desired edit is integrated into the genomic loci by a mechanism that is still not fully understood. Given that scaffold-derived incorporations are much less frequent in mammalian cells, endogenous exonucleases might be involved in this process. Further functional analyses are required to fully understand this mechanism.

strategies to eliminate them will also be crucial for therapeutic applications of prime editing.

Notably, the elongation state (28-nt) revealed that the rearrangement of the PAM-distal target DNA duplex induces the disruption of the base pairs at the end of the guide RNA–target DNA heteroduplex (Extended Data Fig. 10g,h). These observations suggest that further reverse transcription by M-MLV RT leads to additional dissociation between the guide RNA–target DNA heteroduplex, resulting in the detaching of the prime editor from the target site. This might be one of the reasons that the prime editor exhibits low efficiency in inserting long sequences.

Overall, our findings advance researchers' understanding of the intricate mechanism of prime editing, and the structural information will pave the way for the rational engineering of new prime editors with enhanced fidelity and activity.

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

## Methods

### Sample preparation

The PE2 (nSpCas9–engineered M-MLV RT), Cas9 (H840A) and M-MLV RTΔRNaseH (D200N/T306K/W313F/T330P) genes were PCR-amplified from pCMV-PE2 (Addgene plasmid 132775) and assembled separately into pET-based expression vectors with an N-terminal His$_6$-SUMO-tag. The PE6a–d expression plasmids were constructed by replacing the RT gene in the PE2 expression plasmid with the synthesized PE6 RT genes (Eurofins Genomics), respectively. Mutations were introduced by a PCR-based method, and sequences were confirmed by DNA sequencing (Supplementary Table 3). After the plasmids were transformed into *Escherichia coli* Rosetta 2 (DE3), the *E. coli* cells were cultured at 37 °C until the optical density at 600 nm (OD$_{600 \, nm}$) reached 0.8, and protein expression was induced at 20 °C for 18–20 h by the addition of 1 mM isopropyl-β-D-thiogalactopyranoside (Nacalai Tesque). The *E. coli* cells were collected by centrifugation, lysed by sonication in buffer A (20 mM Tris-HCl, pH 8.0, 1 M NaCl and 20 mM imidazole), and clarified by centrifugation. The clarified lysate was incubated with Ni-NTA Superflow resin (Qiagen) at 4 °C for 1 h and loaded into an Econo-Column (Bio-Rad). After the resin was washed with buffer A and buffer B (20 mM Tris-HCl, pH 8.0, 300 mM NaCl and 20 mM imidazole), the protein was eluted with buffer C (20 mM Tris-HCl, pH 8.0, 300 mM NaCl and 300 mM imidazole). The eluted protein was incubated with SUMO protease (produced in-house) at 4 °C overnight, and then loaded onto a HiTrap Heparin column (GE Healthcare) equilibrated with buffer D (20 mM Tris-HCl, pH 8.0 and 300 mM NaCl). The bound protein was eluted with a linear gradient of 0.3–2 M NaCl and further purified on a HiLoad 16/600 Superdex 200 pg column (GE Healthcare) equilibrated with buffer E (20 mM Tris-HCl, pH 8.0, 500 mM NaCl, 2 mM MgCl$_2$ and 1 mM DTT). The peak fractions were collected and stored at −80 °C until use.

### pegRNA preparation

Templates for in vitro transcription were prepared by annealing a forward T7 promoter oligonucleotide with an oligonucleotide containing the reverse complement of the T7 promoter and a pegRNA sequence (Supplementary Table 3). The in vitro transcription reaction was performed at 37 °C overnight, in 50 mM Tris-HCl, pH 8.0, 40 mM KCl, 20 mM MgCl$_2$, 5 mM each NTP, 10 mM GMP, 5 mM DTT, 2 mM spermidine, 1 U ml$^{-1}$ inorganic pyrophosphatase (Sigma), 80 µg ml$^{-1}$ T7 RNA polymerase (produced in-house) and 20 nM template. The transcribed pegRNA was purified by 8% denaturing urea-PAGE, extracted from gel slices with Tris borate–EDTA buffer (Takara) and then ethanol precipitated. The pegRNA pellet was dissolved in nuclease-free water and stored at −20 °C.

### In vitro prime editing assay

All in vitro prime editing reactions were performed using 5′-Cy5-labelled pre-nicked DNA substrates. These DNA substrates were annealed with three oligonucleotides (5′-Cy5-NTS, NTS-3′ and TS; 1:1:1 molar ratio for Fig. 1c and 1:1.5:1 molar ratio for the other experiments) (Supplementary Table 1) by heating to 95 °C for 2 min followed by slowly cooling to room temperature. For the pegRNA-MM, TS-MM was used in place of TS. When using untethered PE in place of PE2 in the reaction, purified dSpCas9 and purified RTΔRH were mixed at a molar ratio of 1:1 and handled like PE2 in the subsequent steps. The PE2–pegRNA complex (1.6 µM or 3.0 µM) was prepared by mixing the purified PE2 and pegRNA at 37 °C for 3 min. The binary complex (5 µl) was mixed with the 5′-Cy5-labelled pre-nicked DNA substrate (5 µl, 200 nM final concentration) and incubated at 37 °C for 10 min in PE reaction buffer (20 mM HEPES-NaOH, pH 7.5, 100 mM NaCl, 5% glycerol, 3 mM MgCl$_2$, 0.2 mM EDTA and 5 mM DTT) supplemented with 250 µM each dNTP or U-Stall Solution (250 µM ddATP, 250 µM dTTP, 250 µM dGTP and 250 µM dCTP). The reaction was stopped by the addition of quench buffer containing EDTA (0.5 mM final concentration) and Proteinase

K (60 ng). Aliquots (2 µl) were mixed with quench buffer (3 µl), and the reaction products were separated on 10% or 15% Novex PAGE TBE–urea gels (Invitrogen) and then visualized using an Amersham Imager 600 (GE Healthcare). The reverse transcription efficiencies of each group were calculated using Image J (ref. 23). In vitro prime editing experiments were performed at least three times.

### Cryo-EM sample preparation

The 51-nt pre-nicked DNA substrates for cryo-EM samples were prepared by annealing three nucleotides (5′-NTS+3nt, NTS-3′ and TS-MM; 1:1:1 molar ratio). For the pre-initiation and initiation complexes, 5′-NTS was used in place of 5′-NTS+3nt (Supplementary Table 2). The dSpCas9–RTΔRH–pegRNA–target DNA complexes were reconstituted by incubating the purified dSpCas9, RTΔRH, the 115-nt or 137-nt pegRNA-MM and the 51-nt pre-nicked DNA substrate at a molar ratio of 6:6:8:3 at 37 °C for 30 min in PE reconstitution buffer (20 mM HEPES-NaOH, pH 7.5, 100 mM NaCl, 2.5% glycerol and 2 mM MgCl$_2$), supplemented with 250 µM ddATP (for the initiation state) or U-Stall Solution (for the other states). The dSpCas9–pegRNA–target DNA complex (the pre-initiation complex) was reconstituted similarly without RTΔRH. The reconstituted complexes were purified by size-exclusion chromatography on a Superdex 200 Increase 10/300 column (GE Healthcare) equilibrated with buffer F (20 mM Tris-HCl, pH 8.0, 150 mM NaCl, 2 mM MgCl$_2$ and 1 mM DTT). The purified complex solution ($A_{260}$ = 4.6–11) was applied to Au 300 mesh R1.2/1.3 grids (Quantifoil), which were freshly glow-discharged with 3 µl amylamine, using a Vitrobot Mark IV (FEI) at 4 °C and 100% humidity, with a waiting time of 10 s and a blotting time of 4 s. The grids were then plunge-frozen in liquid ethane cooled at liquid nitrogen temperature.

### Cryo-EM data collection

Cryo-EM data for the initiation, pre-initiation and elongation (28-nt) complexes were collected using a Titan Krios G3i microscope (Thermo Fisher Scientific) and for the other complexes using a Titan Krios G4 microscope (Thermo Fisher Scientific), both running at 300 kV and equipped with a Gatan Quantum-LS Energy Filter (GIF) and a Gatan K3 Summit direct electron detector in electron counting mode (University of Tokyo). All movies were recorded at a nominal magnification of 105,000×, corresponding to a calibrated pixel size of 0.83 Å, with a total dose of approximately 50 electrons per Å$^2$ per 48 frames. The data were automatically acquired using the EPU software (Thermo Fisher Scientific). The dose-fractionated movies of the pre-initiation and elongation (28-nt) complexes were subjected to beam-induced motion correction and dose weighting using MotionCor2 (ref. 24) in RELION v.3.1.1 (ref. 25); those of the termination and initiation complexes were processed using patch motion correction in cryoSPARC v.3.3.2 (ref. 26); and those of the elongation complex (16-nt) were handled using patch motion correction in cryoSPARC v.4.2.1. The contrast transfer function (CTF) parameters for the termination and initiation complexes, the pre-initiation and elongation (16-nt) complexes and the elongation (28-nt) complex were estimated using patch-based CTF estimation in cryoSPARC versions 3.3.2, 4.2.1 and v4.4, respectively.

### Single-particle cryo-EM data processing

Data for the termination and initiation complexes were processed using cryoSPARC v3.3.2 and v4.2.1. Data for the pre-initiation and elongation (16-nt) complexes and the elongation (28-nt) complex were processed using cryoSPARC v4.2.1 and v4.4, respectively. All reported resolutions are based on the gold-standard Fourier shell correlation with a cut-off of 0.143[27], and the local resolution was estimated with BlocRes[28] in cryoSPARC.

For the termination complex, 1,112,419 particles were selected using a Topaz picking model from the 4,363 motion-corrected and dose-weighted micrographs, and extracted at a pixel size of 3.32 Å. These particles were subjected to two rounds of two-dimensional (2D) classification to separate 671,078 promising particles from junk

particles. Then, 500,000 particles were randomly selected from each particle set, and subsequently used for ab initio reconstruction to generate good initial and junk maps. All of the extracted particles were further curated by three rounds of heterogeneous refinement with two good initial and two junk maps, while updating the two good reference maps. The 248,187 particles in the best class were re-extracted at a pixel size of 1.30 Å and subsequently refined using non-uniform refinement[29] with optimization of the CTF value, resulting in the 3.00-Å overall map. Particle subtraction was performed on the refined particles using a mask around the Cas9–pegRNA scaffold region, and the signal-subtracted particles were used for local refinement (rotation search extent 5 deg, shift search extent 2 Å, initial lowpass resolution 8 Å) with a local mask around the RTΔRH, resulting in the 3.48-Å local map. Finally, the overall and local maps were merged into the final composite map, using the vop maximum command in UCSF ChimeraX[30].

For the initiation complex, 2,532,892 particles were chosen using a Topaz picking model from the 5,266 motion-corrected and dose-weighted micrographs, and extracted at a pixel size of 3.32 Å, as described above. These particles were subjected to two rounds of 2D classification to select 1,607,568 promising particles, which were further curated through three rounds of heterogeneous refinement, as described above. The 656,084 particles in the best class were re-extracted at a pixel size of 1.30 Å and then subjected to 3D classification (five classes, target resolution = 4 Å, PCA initialization mode) with a focus mask around the RTΔRH. The 118,125 particles in the best class were refined using non-uniform refinement, resulting in the 3.12-Å overall map. To further improve the local resolution around the RTΔRH, particle subtraction and local refinement were performed as described above, resulting in the 4.10-Å local map around the RTΔRH. Finally, the overall and local maps were merged into the final composite map, using the vop maximum command in UCSF ChimeraX.

For the pre-initiation complex, 3,357,907 particles were selected using a Topaz picking model from the 8,154 motion-corrected and dose-weighted micrographs, and extracted at a pixel size of 3.32 Å. These particles were subjected to two rounds of 2D classification to select 1,382,881 promising particles, which were further curated through three rounds of heterogeneous refinement in a similar manner to the procedure used for the termination complex. The 976,259 particles in the good classes were re-extracted with a pixel size of 1.30 Å and subsequently refined using non-uniform refinement, resulting in a 3.11-Å map, in which the density for the PBS–NTS heteroduplex was, however, unresolved. Therefore, the aligned particles were subjected to 3D classification (five classes, target resolution = 5 Å, PCA initialization mode) with a focus mask around the position of the PBS–NTS heteroduplex in the initiation complex. The 197,777 particles in the best class were refined using non-uniform refinement with optimization of the CTF value, resulting in the final 3.22-Å overall map.

For the elongation complex (16-nt), 3,208,543 particles were chosen using a Topaz picking model from the 7,932 motion-corrected and dose-weighted micrographs, and extracted at a pixel size of 3.32 Å. These particles were subjected to two rounds of 2D classification to select 2,262,020 promising particles, which were further curated through three rounds of heterogeneous refinement in a similar manner to the procedure used for the termination complex. The 924,985 particles in the two good classes were re-extracted at a pixel size of 1.30 Å and subjected to 3D classification (four classes, target resolution = 6 Å, PCA initialization mode) with a focus mask around the RTΔRH. The 133,711 particles in the best class were then refined with a manually generated solvent mask just before non-uniform refinement with optimization of the CTF value, resulting in the 3.10-Å overall map. To further improve the local resolution around the RTΔRH, particle subtraction and local refinement were performed as described for the termination complex, resulting in the 6.06-Å local map around the RTΔRH. Finally, the overall and local maps were merged into the final composite map, using the vop maximum command in UCSF ChimeraX.

For the elongation complex (28-nt), 4,851,974 particles were selected using a Topaz picking model from the 9,872 motion-corrected and dose-weighted micrographs, and extracted at a pixel size of 3.32 Å. These particles were subjected to two rounds of 2D classification to select 2,800,847 promising particles, which were further curated through three rounds of heterogeneous refinement in a similar manner to the procedure used for the termination complex. The 702,552 particles in the best class were re-extracted at a pixel size of 1.15 Å and subjected to 3D classification (six classes, target resolution = 4 Å, PCA initialization mode) with a focus mask around the RTΔRH and the RNA–DNA heteroduplex along with Cas9. The 104,057 particles in the best class were refined using non-uniform refinement with optimization of the CTF value, resulting in the final 3.19-Å overall map. To further improve the local resolution around the RTΔRH, particle subtraction and local refinement were performed as described for the termination complex, resulting in the 4.54-Å local map around the RTΔRH. Finally, the overall and local maps were merged into the final composite map, using the vop maximum command in UCSF ChimeraX.

## Model building and validation

The model of the termination complex was built using the cryo-EM structure of the SpCas9–sgRNA–target DNA complex in the checkpoint state (PDB 7Z4L; ref. 12) and the crystal structure of apo-M-MLV RT (PDB 4MH8; ref. 17) as the reference models, followed by manual model building using Coot (ref. 31) against the final density map sharpened using DeepEMhancer. The models of the other complexes were built using the model of the termination complex as the reference, followed by manual model building using Coot and ISOLDE (ref. 32) against the final density map sharpened using DeepEMhancer or local-resolution filtering in cryoSPARC. All models were refined using phenix.real_space_refine v.1.20.1 (ref. 33) with secondary structure and base pair restraints. The structure validation was performed using MolProbity in the PHENIX package[34]. The EMRinger score[35] and 3DFSC sphericity[36] were calculated by PHENIX and by the 3DFSC Processing Server (https://3dfsc.salk.edu/upload/info/), respectively. The statistics of the 3D reconstruction and model refinement are summarized in Extended Data Table 1. The cryo-EM density map figures were generated using UCSF ChimeraX. Molecular graphics figures were prepared using UCSF ChimeraX and CueMol (http://www.cuemol.org).

## Mammalian prime editing assay

HEK293FT cells were purchased from Thermo Fisher Scientific (R70007) and maintained in DMEM-GlutaMAX (Thermo Fisher Scientific, 10569044) with 1× penicillin–streptomycin (Thermo Fisher Scientific, 15140122) and 10% FBS (VWR, 97068-085) at 37 °C with 5% $CO_2$. The cells were seeded at a density of $2 \times 10^4$ cells per well in 96-well plates for transfection. Transfections were performed using Lipofectamine 3000 (Thermo Fisher Scientific, L3000015) when cells reached around 90% confluency. In total, 200 ng plasmids, including 150 ng PE plasmid with 50 ng pegRNA plasmid for PE2, or 135 ng PE plasmid and 50 ng pegRNA with 15 ng sgRNA plasmid for PE3, were transfected into each well. Three wells were transfected for each condition. Three days after transfection, genomic DNA was extracted using 50 μl QuickExtract DNA extraction solution (Lucigen, QE09050) by cycling at 65 °C for 15 min, 68 °C for 15 min and 95 °C for 10 min. Two rounds of PCR were conducted to amplify target sites with NEBNext High-Fidelity 2× PCR Master Mix (NEB, M0541L). For the first round of PCR, 2.5 μl of cell lysate was used as the template in 10-μl PCR reactions under the following thermal cycling conditions: one cycle, 98 °C, 30 s; 12 cycles, 98 °C, 10 s, 69 °C, 20 s, 72 °C, 30 s; one cycle, 72 °C, 2 min; 4 °C hold. For the second round of PCR, 1 μl of PCR product from the first round was used as the template in 10-μl PCR reactions under the following thermal cycling conditions: one cycle, 98 °C, 30 s; 18 cycles, 98 °C, 10 s, 63 °C, 20 s, 72 °C, 30 s; one cycle, 72 °C, 5 min; 4 °C hold. All amplicons were sequenced using a MiSeq Reagent Kit v.2, 300-cycle

(Illumina, MS-102-2002). The prime editing efficiency was quantified using the published CRISPResso2 pipeline[37].

## Reporting summary

Further information on research design is available in the Nature Portfolio Reporting Summary linked to this article.

## Data availability

The atomic models have been deposited in the PDB under the accession codes 8WUS (termination state), 8WUT (initiation state), 8WUU (pre-initiation state), 8WUV (elongation (16-nt) state) and 8YGJ (elongation (28-nt) state). The cryo-EM density map has been deposited in the Electron Microscopy Data Bank under the accession codes EMD-37858 (termination state), EMD-37859 (initiation state), EMD-37860 (pre-initiation state), EMD-37861 (elongation (16-nt) state) and EMD-39253 (elongation (28-nt) state). The next-generation sequencing data have been deposited in the NCBI under accession code PRJNA1084104. All data are available in this Article or its Supplementary Information.

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

**Acknowledgements** We thank R. Nagamura and K. Kobataka for assistance with electron microscopy, and the members of the O.N. laboratory for comments and discussions. Electron microscopy data were collected at the cryo-EM facility at the University of Tokyo. F.Z. is supported by the Howard Hughes Medical Institute; the Hock E. Tan and K. Lisa Yang Center for Autism Research at MIT; the Broad Institute Programmable Therapeutics Gift Donors; the Pershing Square Foundation; William Ackman and Neri Oxman; and the Asness Family Foundation. O.N. was supported by AMED grant numbers JP23fa627001 and JP19am0401005; the Platform Project for Supporting Drug Discovery and Life Science Research (Basis for Supporting Innovative Drug Discovery and Life Science Research (BINDS)) from AMED under grant numbers JP23ama121002 (support number 3272) and JP23ama121012; and the Cabinet Office, Government of Japan, Public/Private R&D Investment Strategic Expansion Program (PRISM), grant number JPJ008000.

**Author contributions** Y.S. performed biochemical and structural analyses with assistance from R.N., M.H., S.N.O., H.H. and Y.I. Y.S. and R.N. performed model building and structural refinement. S.Z. performed cell biological experiments. F.Z. conceived the project. Y.S. and R.N. wrote the manuscript with help from all authors. R.N. and O.N. supervised the research.

**Competing interests** F.Z. is a co-founder of Editas Medicine, Beam Therapeutics, Pairwise Plants, Arbor Biotechnologies, Sherlock Biosciences and Aera Therapeutics. F.Z. is a scientific advisor for Octant. O.N. is a co-founder of, board member of and scientific advisor for Curreio.

**Additional information**
**Correspondence and requests for materials** should be addressed to Ryoya Nakagawa or Osamu Nureki.

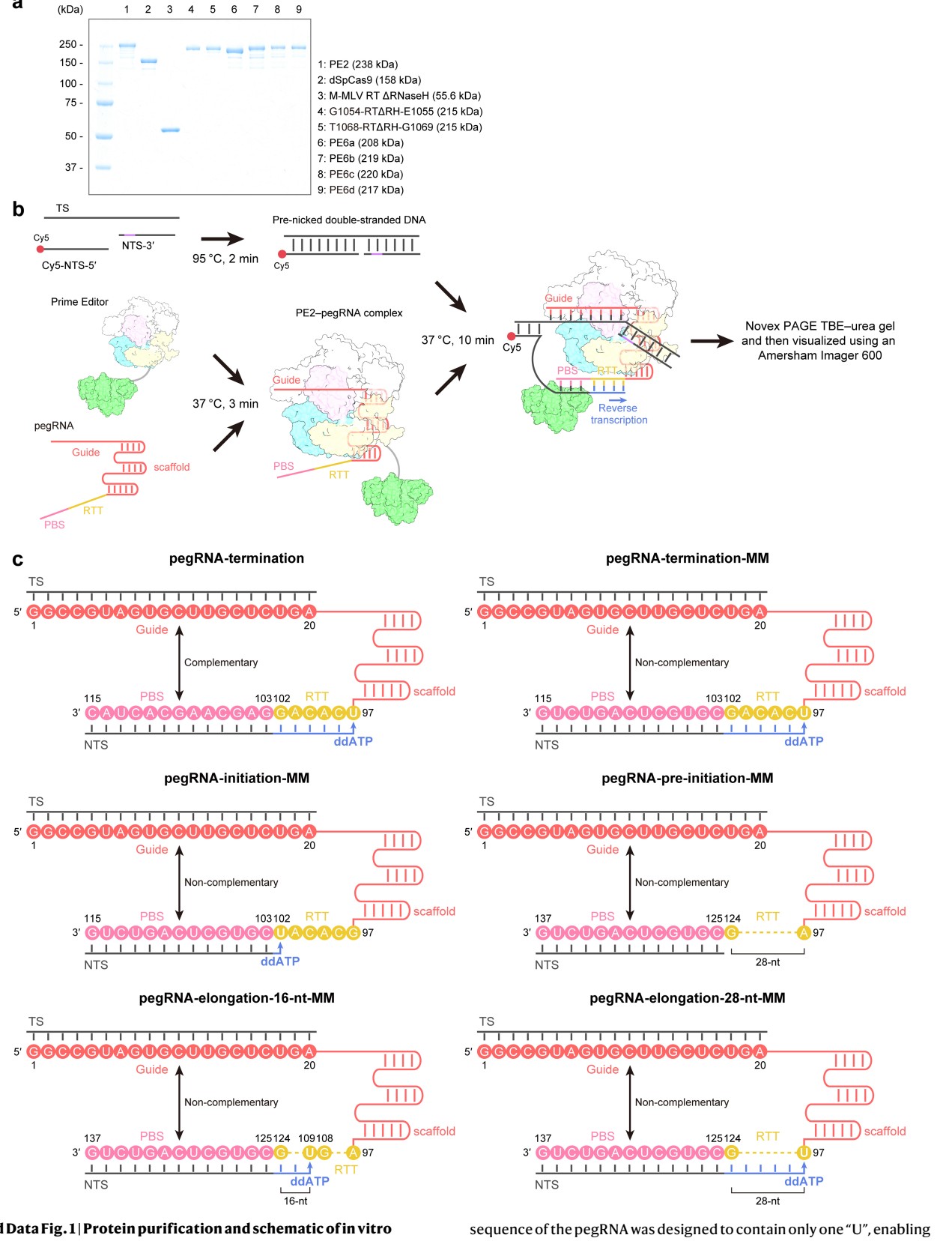

**Extended Data Fig. 1 | Protein purification and schematic of in vitro prime editing assay. a**, SDS-PAGE gel of purified PE2, dSpCas9, RTΔRH, PE2 mutants, and PE6a–d proteins. **b**, Schematic of the in vitro prime editing assay. The pegRNA and DNA sequences used in the assay are summarized in Supplementary Table 1. **c**, The pegRNA sequences used for the in vitro prime editing assay and structural determinations in multiple states. The RTT sequence of the pegRNA was designed to contain only one "U", enabling reverse transcription to stop at this "U" using ddATP. Recent studies showed that the complementarity between the guide sequence and PBS sequence in the pegRNA inhibits the prime editing activity[9,10]. Thus, we performed the structural analysis using a pegRNA with non-complementary guide and PBS sequences.

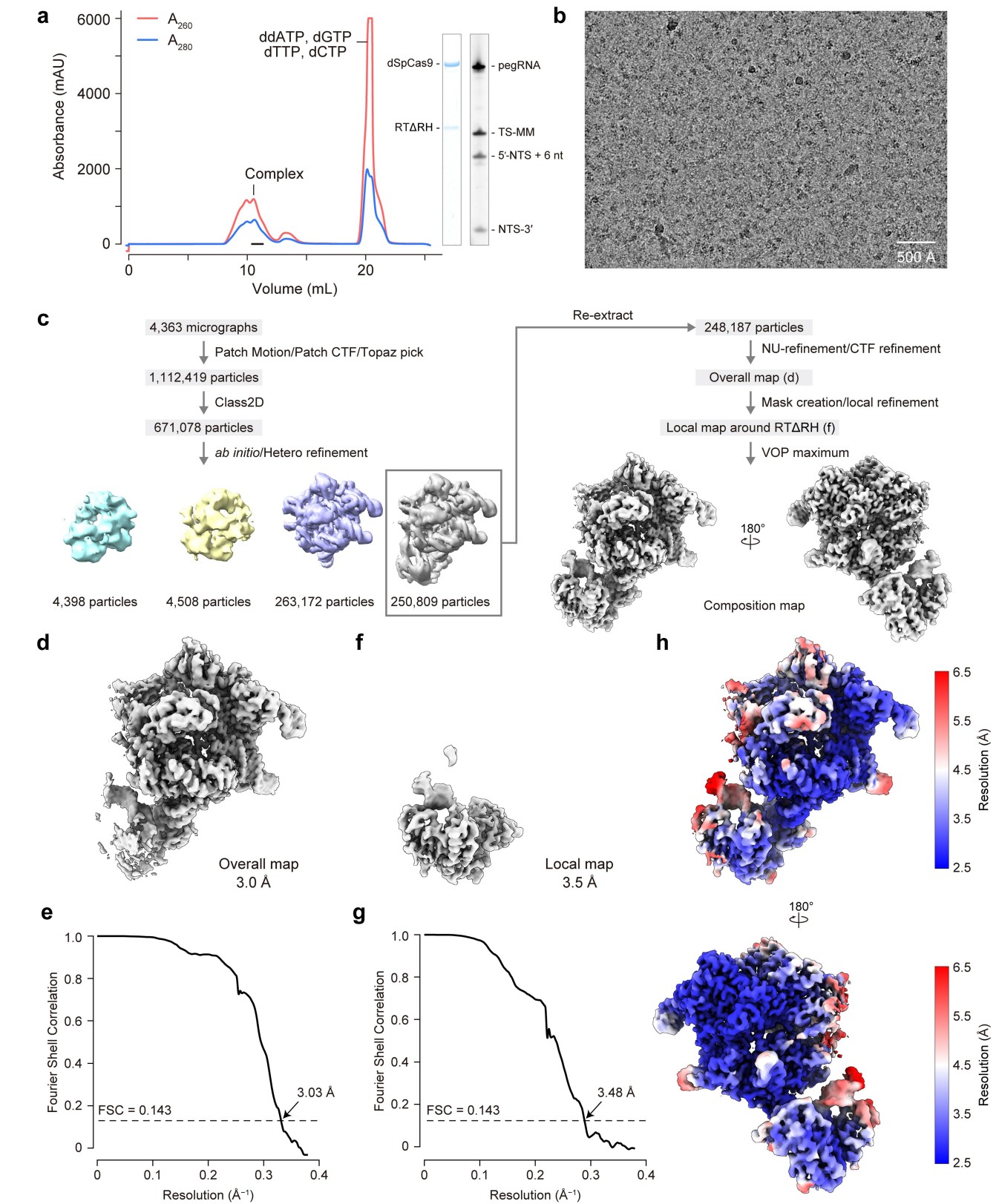

**Extended Data Fig. 2 | Single-particle cryo-EM analysis of the termination state. a**, Size-exclusion chromatography profile of the SpCas9–RTΔRH–pegRNA–target DNA complex in the termination state. The peak fraction (indicated by a black bar) was analysed by SDS-PAGE and urea-PAGE, and then used for cryo-EM analysis. **b**, Representative cryo-EM image of the SpCas9–RTΔRH–pegRNA–target DNA complex in the termination state, recorded on a 300 kV Titan Krios microscope with a K3 camera. **c**, Single-particle cryo-EM image processing workflow. **d,f**, Overall (**d**) and local (**f**) cryo-EM density maps. **e,g**, Fourier shell correlation (FSC) curves for the overall (**e**) and local (**g**) 3D reconstructions. The gold-standard cut-off (FSC = 0.143) is marked with the black dotted line. **h**, Local-resolution cryo-EM density map.

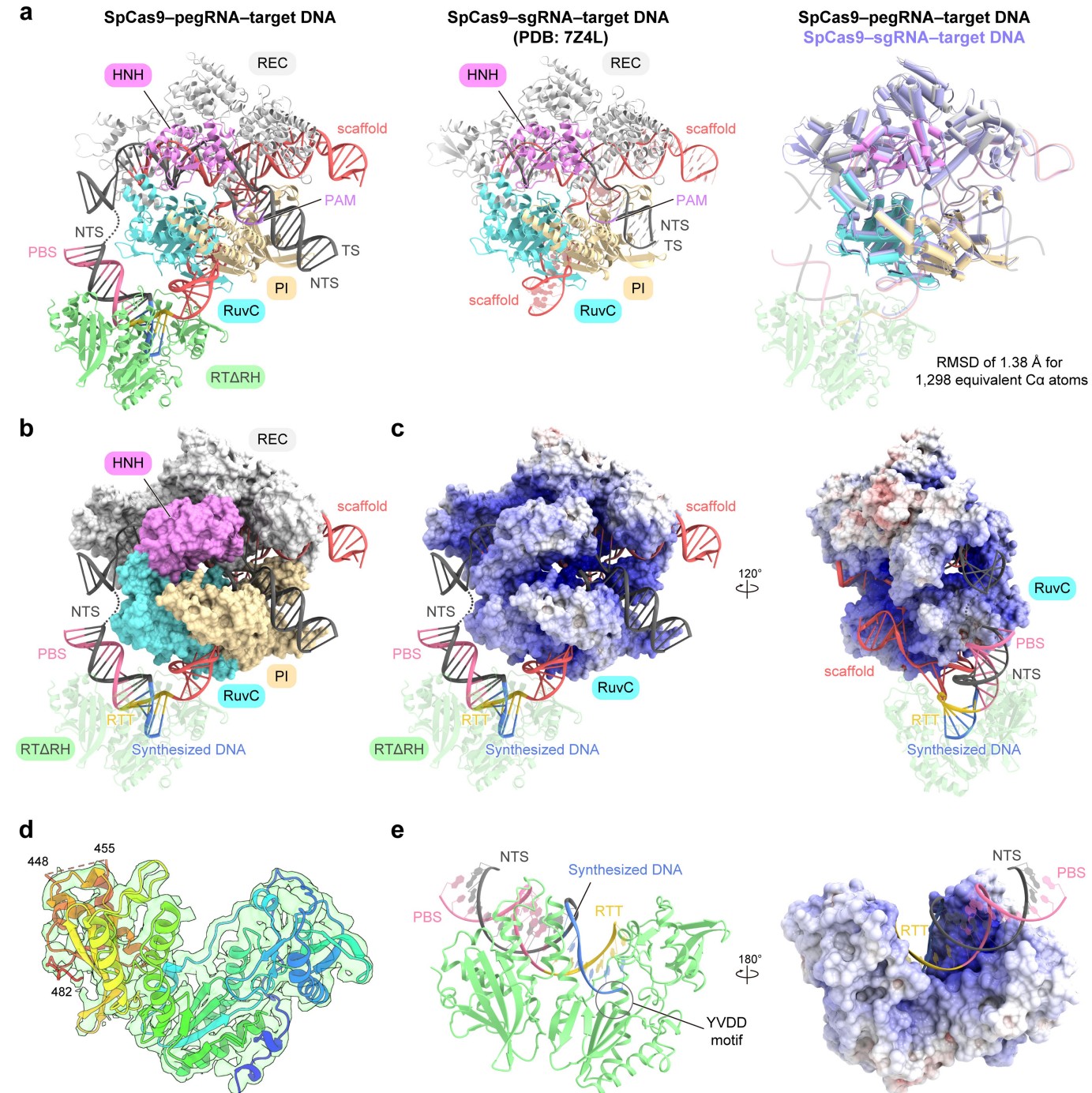

**Extended Data Fig. 3 | Overall structure of the prime editor in the termination state. a**, Structural comparison of the SpCas9–pegRNA–target DNA complex with the SpCas9–sgRNA–target DNA complex (PDB: 7Z4L)[12]. The SpCas9–sgRNA–target DNA complex (light blue) is superimposed onto the SpCas9–pegRNA–target DNA complex. **b**,**c**, Surface representation (**b**) and electrostatic surface potential (**c**) of SpCas9 in the termination state. The PBS–NTS heteroduplex is located on the weakly positively charged surface facing

the RuvC domain. The disordered regions are indicated as dotted lines. **d**, Ribbon representation of RTΔRH with cryo-EM density. Residues 1–23, 449–454, and 483–496 are disordered due to their flexibilities. **e**, Ribbon representation (left) and electrostatic surface potential (right) of RTΔRH with the PBS–NTS and RTT–synthesized DNA heteroduplexes. These heteroduplexes are accommodated within the positively charged groove of the RTΔRH. The catalytic YVDD motif is indicated by the grey circle.

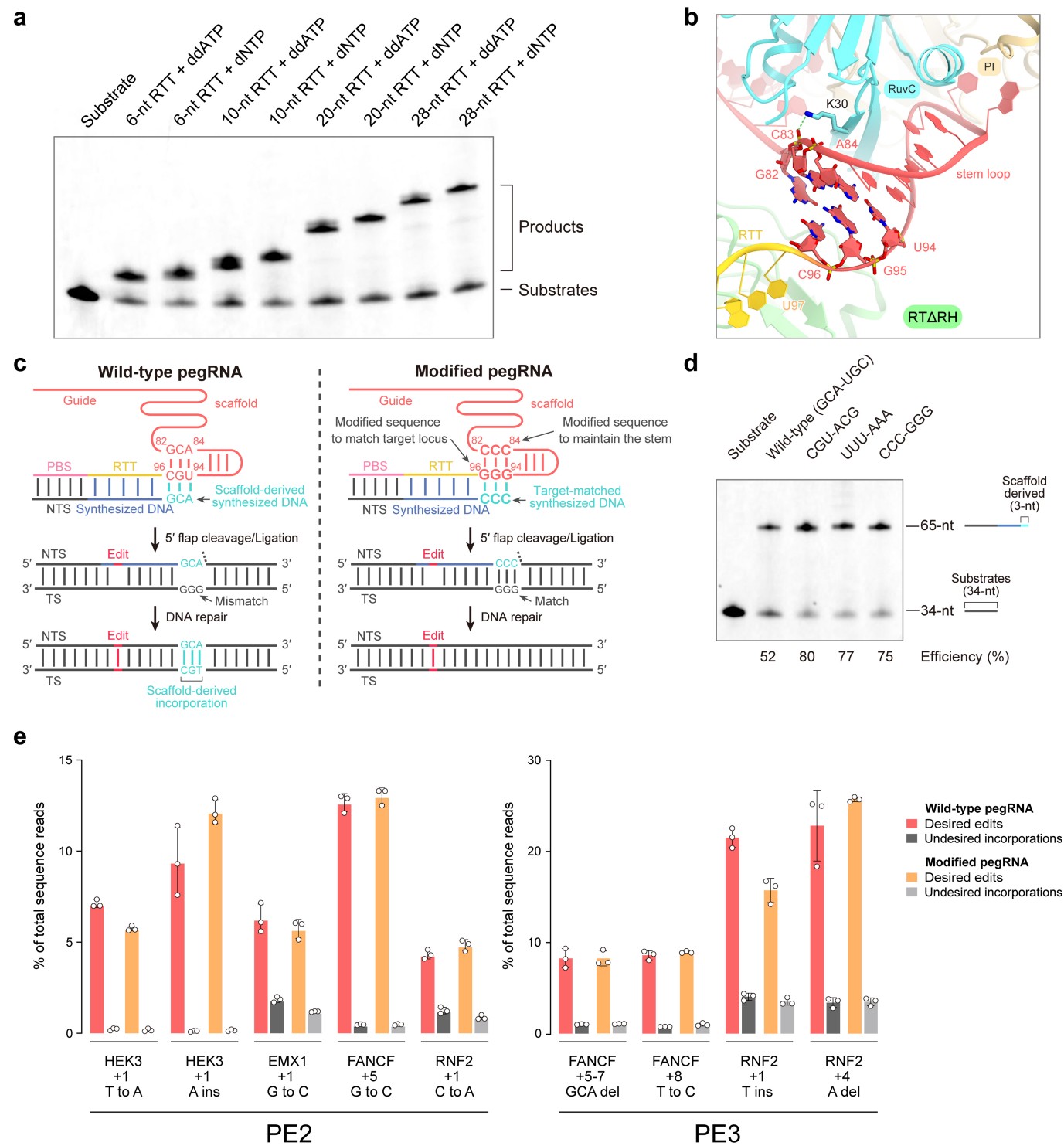

**Extended Data Fig. 4 | pegRNA modification. a**, In vitro prime editing assay using pegRNAs containing 6-, 10-, 20-, and 28-nt RTT sequences. The PE2–pegRNA complexes were added to DNA substrates with dNTPs or with dCTP, dTTP, dGTP, and ddATP (referred to as ddATP for simplicity), and incubated at 37 °C for 10 min. The reaction products were separated on a 10% Novex PAGE TBE–urea gel, and then Cy5 fluorescence was visualized. The experiments were repeated three times with similar results. **b**, Close-up view of the 3′ stem loop in the pegRNA scaffold. Three base pairs (G82-C96–A84-U94) do not form base-specific interactions with SpCas9, while the phosphate backbone of C93 forms hydrogen bonds with K30. The hydrogen bond is depicted with a green dashed line. **c**, Schematic of the wild-type and modified pegRNAs. The modified pegRNA has an altered U94–C96 to complement the target locus and adjust A84–G82 to maintain the stem structure. **d**, In vitro prime editing assay using the wild-type pegRNA and three modified pegRNAs. The three modified pegRNAs were altered from GCA (82–84)-UGC (94–96) to CGU-ACG, UUU-AAA, and CCC-GGG, respectively. The experiments were repeated three times with similar results. **e**, The desired edit and the undesired incorporation efficiencies of the PE2 with five conditions (left) and the PE3 with four conditions (right), using the wild-type and modified pegRNAs in HEK293 cells. Data are mean ± s.d. (n = 3 biologically independent samples).

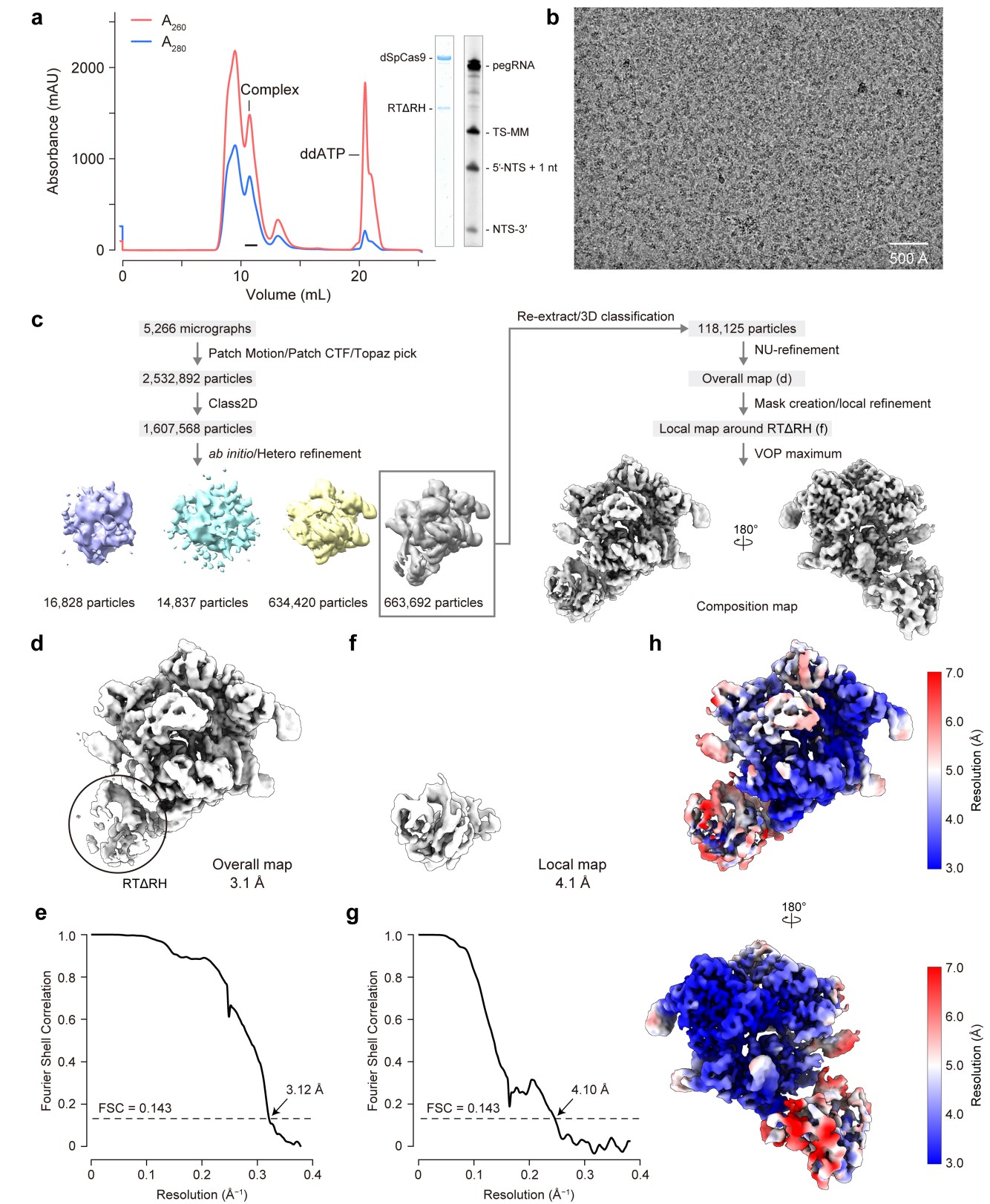

**Extended Data Fig. 5 | Single-particle cryo-EM analysis of the initiation state.**
**a**, Size-exclusion chromatography profile of the SpCas9–RTΔRH–pegRNA–target DNA complex in the initiation state. The peak fraction (indicated by a black bar) was analysed by SDS-PAGE and urea-PAGE, and then used for cryo-EM analysis. **b**, Representative cryo-EM image of the SpCas9–RTΔRH–pegRNA–target DNA complex in the initiation state, recorded on a 300 kV Titan Krios microscope with a K3 camera. **c**, Single-particle cryo-EM image processing workflow. **d,f**, Overall (**d**) and local (**f**) cryo-EM density maps. The ambiguous density corresponding to RTΔRH is enclosed within a black circle in the overall map. **e,g**, Fourier shell correlation (FSC) curves for the overall (**e**) and local (**g**) 3D reconstructions. The gold-standard cut-off (FSC = 0.143) is marked with the black dotted line. **h**, Local-resolution cryo-EM density map.

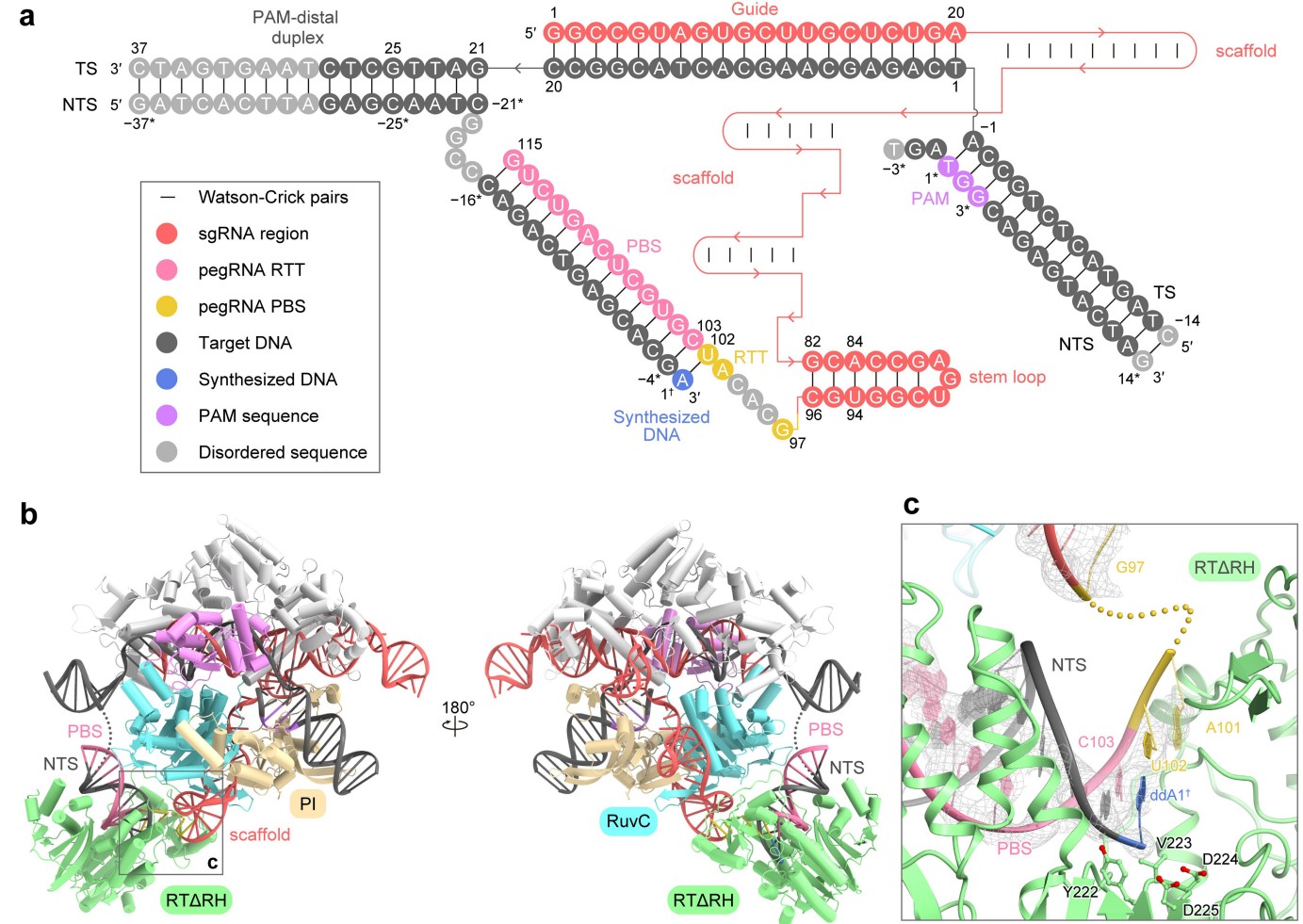

**Extended Data Fig. 6 | Structure of the prime editor in the initiation state.**
**a**, Schematic of the pegRNA and target DNA in the initiation state. Except for the 3′ stem loop (G82–C96), the scaffold region of the pegRNA is represented with a red line for simplicity. The disordered regions are coloured grey. **b**, Overall structure of the SpCas9–RTΔRH–pegRNA–target DNA complex in the initiation state. The disordered regions are indicated as dotted lines. **c**, Close-up view of the M-MLV RT active site in the initiation state. The cryo-EM densities for the RTT, the synthesized DNA, the PBS–NTS heteroduplex, and the 3′ end of the stem loop are shown as grey meshes. The middle of the RTT region (C98–C100) is disordered due to its flexibility. The disordered regions are indicated as dotted lines.

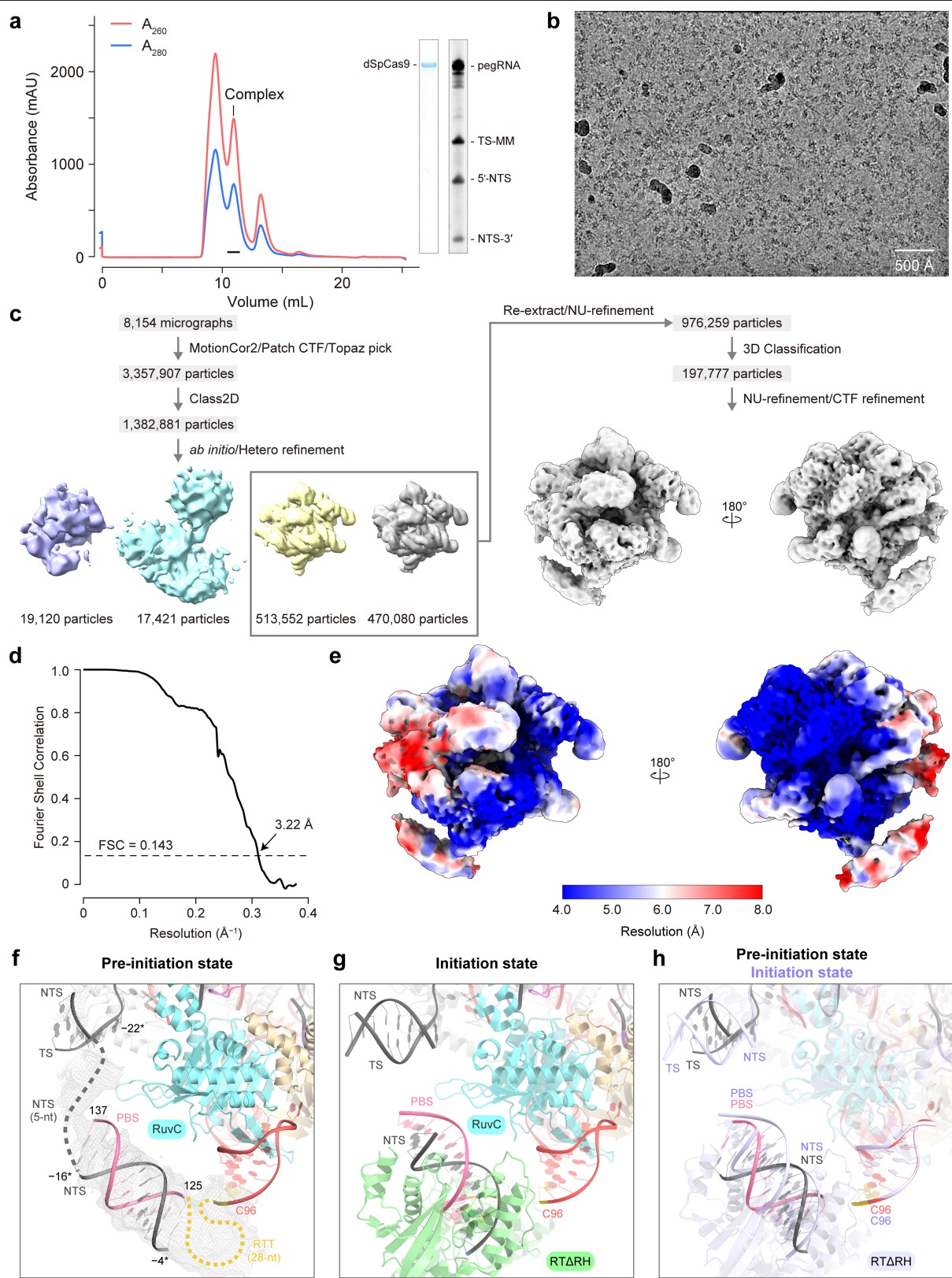

**Extended Data Fig. 7 |** See next page for caption.

**Extended Data Fig. 7 | Structure of the prime editor in the pre-initiation state. a**, Size-exclusion chromatography profile of the SpCas9–pegRNA–target DNA complex in the pre-initiation state. The peak fraction (indicated by a black bar) was analysed by SDS-PAGE and urea-PAGE, and then used for cryo-EM analysis. **b**, Representative cryo-EM image of the SpCas9–pegRNA–target DNA complex in the pre-initiation state, recorded on a 300 kV Titan Krios microscope with a K3 camera. **c**, Single-particle cryo-EM image processing workflow. **d**, Fourier shell correlation (FSC) curves for the 3D reconstruction. The gold-standard cut-off (FSC = 0.143) is marked with the black dotted line. **e**, Local-resolution cryo-EM density map. **f**, Close-up view of the PBS–NTS heteroduplex in the pre-initiation state. The cryo-EM densities for the 3′ extension region of the pegRNA and the NTS are shown as grey meshes. The ambiguous densities corresponding to the single-stranded region of the NTS (dG[−20*]–dC[−17*]) and the RTT (A97–G124) are visible, but we were unable to build the model. **g**, Close-up view of the PBS–NTS heteroduplex in the initiation state. **h**, Comparison of the positions of the PBS–NTS heteroduplex between the pre-initiation and initiation states indicates a similar location in both states. The initiation complex (light blue) is superimposed onto the pre-initiation complex.

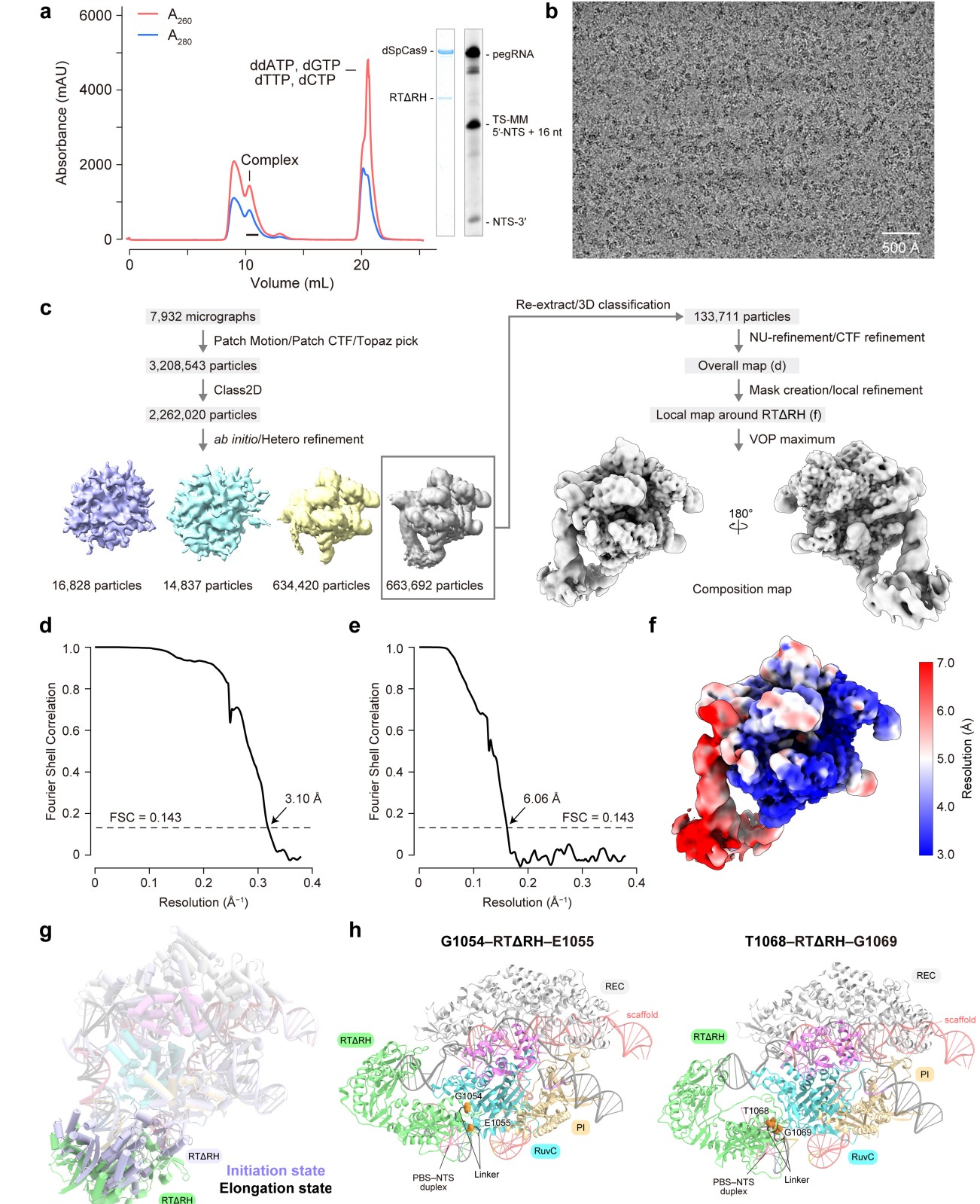

**Extended Data Fig. 8 |** See next page for caption.

**Extended Data Fig. 8 | Structure of the prime editor in the elongation (16-nt) state. a**, Size-exclusion chromatography profile of the SpCas9–RTΔRH–pegRNA–target DNA complex in the elongation (16-nt) state. The peak fraction (indicated by a black bar) was analysed by SDS-PAGE and urea-PAGE, and then used for cryo-EM analysis. **b**, Representative cryo-EM image of the SpCas9–RTΔRH–pegRNA–target DNA complex in the elongation (16-nt) state, recorded on a 300 kV Titan Krios microscope with a K3 camera. **c**, Single-particle cryo-EM image processing workflow. **d,e**, Fourier shell correlation (FSC) curves for the overall (**d**) and local (**e**) 3D reconstructions. The gold-standard cut-off (FSC = 0.143) is marked with the black dotted line. **f**, Local-resolution cryo-EM density map. **g**, Comparison of the RTΔRH positions between the initiation and elongation states. The initiation complex (light blue) is superimposed onto the pre-initiation complex. **h**, The AlphaFold-prediction models of G1054–RTΔRH–E1055 (left) and T1068–RTΔRH–G1069 (right) PE2 variants[38]. The nucleic acid of the initiation state was mapped onto these AlphaFold models. G1054, E1055, T1068, and G1069 are shown as orange space-filling models. The five amino-acid linkers are coloured black.

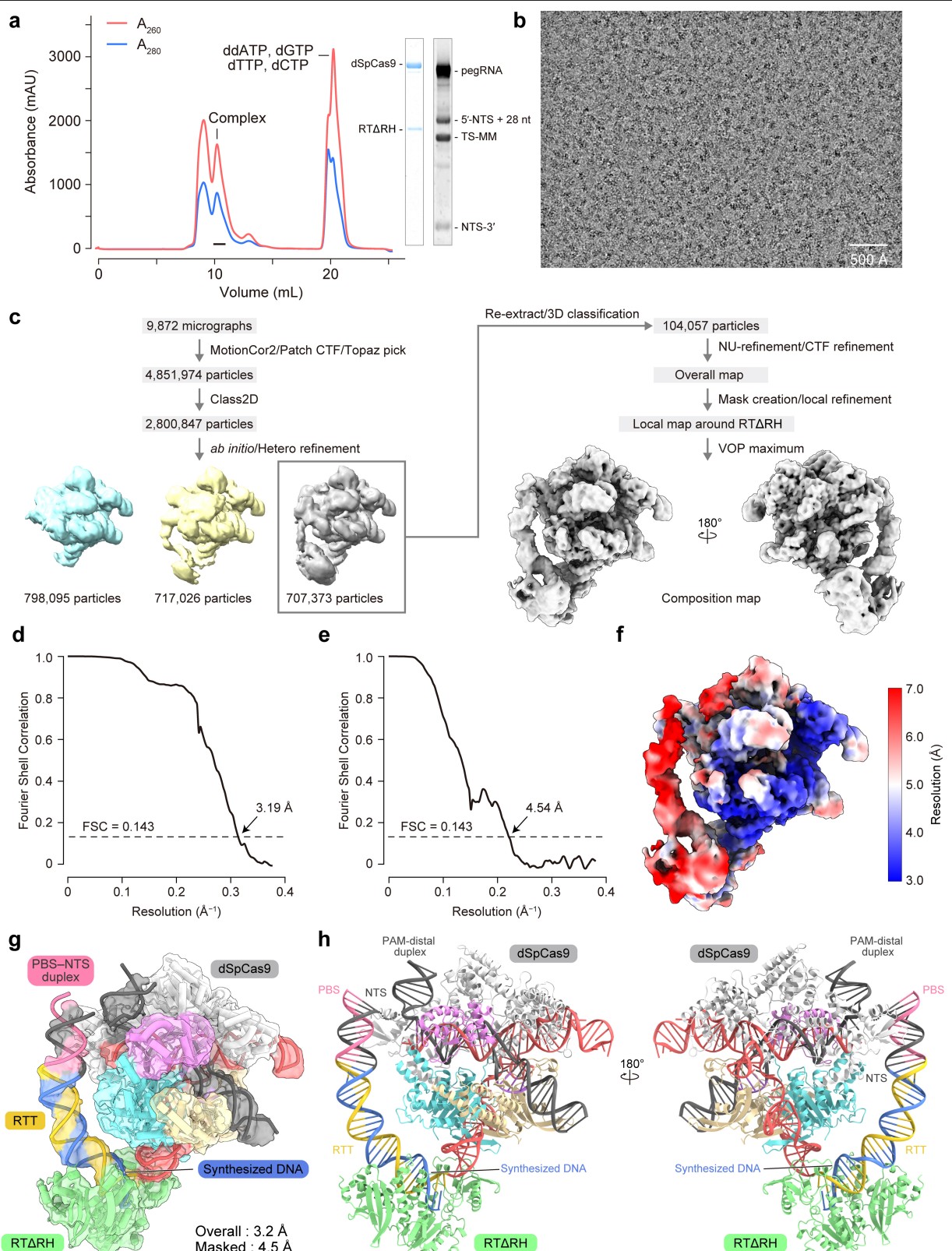

**Extended Data Fig. 9 | Structure of the prime editor in the elongation (28-nt) state. a**, Size-exclusion chromatography profile of the SpCas9–RTΔRH–pegRNA–target DNA complex in the elongation (28-nt) state. The peak fraction (indicated by a black bar) was analysed by SDS-PAGE and urea-PAGE, and then used for cryo-EM analysis. **b**, Representative cryo-EM image of the SpCas9–RTΔRH–pegRNA–target DNA complex in the elongation (28-nt) state, recorded on a 300 kV Titan Krios microscope with a K3 camera. **c**, Single-particle cryo-EM image processing workflow. **d,e**, Fourier shell correlation (FSC) curves for the overall (**d**) and local (**e**) 3D reconstructions. The gold-standard cut-off (FSC = 0.143) is marked with the black dotted line. **f**, Local-resolution cryo-EM density map. **g,h**, The cryo-EM density (**g**) and overall structures (**h**) of the SpCas9–RTΔRH–pegRNA–target DNA complex in the elongation (28-nt) state. The disordered regions are indicated as dotted lines.

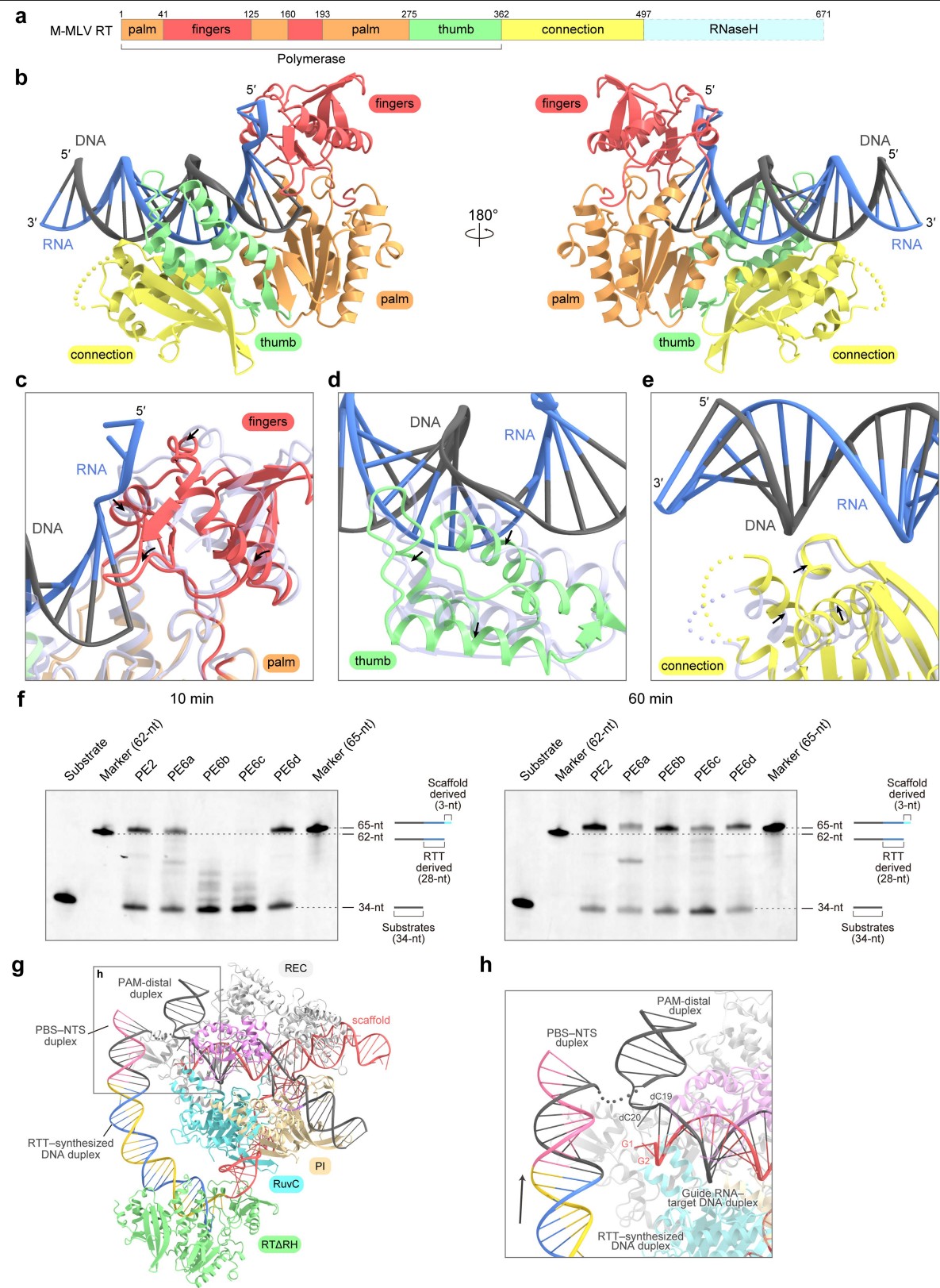

**Extended Data Fig. 10** | See next page for caption.

**Extended Data Fig. 10 | Structure of M-MLV RT with substrates. a**, Domain structure of M-MLV RT. Our construct lacks the RNaseH domain (residues 498–671). **b**, Ribbon representation of M-MLV RT with the RNA–DNA substrate. The disordered regions are indicated as dotted lines. **c–e**, Structural comparison between the substrate-bound and unbound states (PDB: 4MH8)[17] in the fingers (**c**), thumb (**d**) and connection (**e**) domains. The substrate-unbound state (light blue) is superimposed onto the substrate-bound state. **f**, In vitro prime editing assay using PE2 and PE6a–d proteins and the pegRNA containing the 28-nt RTT sequence. The PE2/PE6a–d–pegRNA complex was added to DNA substrates with dNTPs and incubated at 37 °C for 10 min (left) and 60 min (right). The 62-nt marker represents the product transcribed up to the end of the RTT (U97), while the 65-nt marker represents the product transcribed up to U94 in the pegRNA scaffold. The experiments were repeated three times with similar results. **g**, Overall structure of the SpCas9–RTΔRH–pegRNA–target DNA complex in the elongation (28-nt) state. The disordered regions are indicated as dotted lines. **h**, Close-up views of the PBS–NTS heteroduplex, the PAM-distal duplex, and the end of the guide RNA–target DNA heteroduplex. As the reverse transcription of the RTT sequence progresses, the PBS–NTS heteroduplex is pushed in the opposite direction from the RTΔRH (shown in black arrow), resulting in the rearrangement of the PAM-distal duplex. This rearrangement induces the disruption of the base pairs at the end of the guide RNA–target DNA heteroduplex (G1-dC20 and G2-dC19).

**Extended Data Table 1 | Cryo-EM data collection and refinement statistics**

| | Termination (PDB: 8WUS) (EMD-37858) | Initiation (PDB: 8WUT) (EMD-37859) | Pre-initiation (PDB: 8WUU) (EMD-37860) | Elongation 16-nt (PDB: 8WUV) (EMD-37861) | Elongation 28-nt (PDB: 8YGJ) (EMD-39253) |
|---|---|---|---|---|---|
| **Data collection and Processing** | | | | | |
| Microscope | Titan Krios G4 | Titan Krios G3i | Titan Krios G3i | Titan Krios G4 | Titan Krios G3i |
| Detector | Gatan K3 camera | Gatan K3 camera | Gatan K3 camera | Gatan K3 camera | Gatan K3 camera |
| Magnification | 105,000 | 105,000 | 105,000 | 105,000 | 105,000 |
| Voltage (kV) | 300 | 300 | 300 | 300 | 300 |
| Total electron exposure (e$^-$/Å²) | 50 | 50 | 50 | 50 | 50 |
| Defocus range (μm) | −0.8 to −1.6 | −0.8 to −1.6 | −0.8 to −1.6 | −0.8 to −1.6 | −0.8 to −1.6 |
| Pixel size (Å) | 0.83 | 0.83 | 0.83 | 0.83 | 0.83 |
| Symmetry imposed | $C_1$ | $C_1$ | $C_1$ | $C_1$ | $C_1$ |
| Number of movies | 4,363 | 5,266 | 8,154 | 7,932 | 9,872 |
| Initial particles | 1,112,419 | 2,532,892 | 3,357,907 | 3,208,543 | 4,851,974 |
| Final particles | 248,187 | 118,125 | 197,777 | 133,711 | 104,057 |
| Resolution (global, Å) FSC 0.143 (masked) (Overall/ Local) | 3.00/ 3.48 | 3.12/ 4.10 | 3.22/ − | 3.10/ 6.06 | 3.19/ 4.54 |
| Map-sharpening $B$ factor (Å²) | −60 | −60 | −60 | −60 | −60 |
| **Model composition** | | | | | |
| Protein atoms | 14,322 | 14,322 | 10,751 | 14,421 | 14,412 |
| Nucleic acid atoms | 4,037 | 3,994 | 3,812 | 4,479 | 5,061 |
| **Model Refinement** | | | | | |
| Model-Map CC ($CC_{mask}$/ $CC_{box}$/ $CC_{peaks}$/ $CC_{volume}$) | 0.74/ 0.77/ 0.68/ 0.72 | 0.67/ 0.79/ 0.64/ 0.65 | 0.59/ 0.77/ 0.56/ 0.83 | 0.75/ 0.76/ 0.67/ 0.72 | 0.79/ 0.84/ 0.71/ 0.77 |
| Resolution (Å) by model-to-map FSC, threshold 0.50 (masked/ unmasked) | 3.0/ 3.0 | 3.2/ 3.3 | 3.4/ 3.4 | 3.2/ 3.2 | 3.2/ 3.2 |
| Average $B$ factor (Å²) (protein/ nucleotides) | 204.04/ 263.77 | 478.04/ 829.22 | 138.61/ 7033.91 | 98.42/ 139.12 | 98.80/ 173.81 |
| R.M.S. deviations | | | | | |
| bond lengths (Å) | 0.002 | 0.005 | 0.006 | 0.003 | 0.003 |
| bond angles (°) | 0.419 | 0.545 | 0.581 | 0.467 | 0.555 |
| **Validation** | | | | | |
| MolProbity score | 1.38 | 1.50 | 1.44 | 1.42 | 1.31 |
| CaBLAM outliers (%) | 1.27 | 1.44 | 1.31 | 1.14 | 1.14 |
| Clash score | 6.96 | 8.09 | 6.51 | 7.27 | 4.08 |
| Rotamer outliers (%) | 0.00 | 0.00 | 0.00 | 0.00 | 0.00 |
| Cβ outliers (%) | 0.00 | 0.00 | 0.00 | 0.00 | 0.00 |
| EMRinger score | 2.55 | 1.73 | 2.31 | 2.13 | 2.25 |
| 3DFSC sphericity | 0.957 | 0.869 | 0.860 | 0.950 | 0.858 |
| Ramachandran plot | | | | | |
| Favored (%) | 98.23 | 97.72 | 97.62 | 97.90 | 97.39 |
| Allowed (%) | 1.71 | 2.23 | 2.38 | 1.98 | 2.61 |
| Outlier (%) | 0.06 | 0.06 | 0.00 | 0.11 | 0.00 |

The structure validation was performed using MolProbity in the PHENIX package. The EMRinger score and 3DFSC sphericity were calculated by PHENIX and the 3DFSC processing Server (https://3dfsc.salk.edu/upload/info/), respectively.

# Reporting Summary

## Statistics

For all statistical analyses, confirm that the following items are present in the figure legend, table legend, main text, or Methods section.

| n/a | Confirmed | |
|---|---|---|
| ☐ | ☒ | The exact sample size ($n$) for each experimental group/condition, given as a discrete number and unit of measurement |
| ☐ | ☒ | A statement on whether measurements were taken from distinct samples or whether the same sample was measured repeatedly |
| ☒ | ☐ | The statistical test(s) used AND whether they are one- or two-sided<br>*Only common tests should be described solely by name; describe more complex techniques in the Methods section.* |
| ☒ | ☐ | A description of all covariates tested |
| ☒ | ☐ | A description of any assumptions or corrections, such as tests of normality and adjustment for multiple comparisons |
| ☐ | ☒ | A full description of the statistical parameters including central tendency (e.g. means) or other basic estimates (e.g. regression coefficient) AND variation (e.g. standard deviation) or associated estimates of uncertainty (e.g. confidence intervals) |
| ☒ | ☐ | For null hypothesis testing, the test statistic (e.g. $F$, $t$, $r$) with confidence intervals, effect sizes, degrees of freedom and $P$ value noted<br>*Give P values as exact values whenever suitable.* |
| ☒ | ☐ | For Bayesian analysis, information on the choice of priors and Markov chain Monte Carlo settings |
| ☒ | ☐ | For hierarchical and complex designs, identification of the appropriate level for tests and full reporting of outcomes |
| ☒ | ☐ | Estimates of effect sizes (e.g. Cohen's $d$, Pearson's $r$), indicating how they were calculated |

*Our web collection on statistics for biologists contains articles on many of the points above.*

## Software and code

Policy information about availability of computer code

| Data collection | EPU (version 2.12) |
|---|---|
| Data analysis | RELION-3.1.1, cryoSPARC (v3.3.2, v4.2.1 and v4.4), COOT (version 0.9), ISOLDE, DeepEMhancer, UCSF ChimeraX (version 1.1.1 and 1.6.1 ), CueMol2 (http://www.cuemol.org/ version 2.2.3.443), PHENIX (version 1.20.1), ImageJ (version 2.14.0), and CRISPResso2 |

For manuscripts utilizing custom algorithms or software that are central to the research but not yet described in published literature, software must be made available to editors and reviewers. We strongly encourage code deposition in a community repository (e.g. GitHub). See the Nature Portfolio guidelines for submitting code & software for further information.

## Data

Policy information about availability of data

All manuscripts must include a data availability statement. This statement should provide the following information, where applicable:
- Accession codes, unique identifiers, or web links for publicly available datasets
- A description of any restrictions on data availability
- For clinical datasets or third party data, please ensure that the statement adheres to our policy

The atomic models have been deposited in the Protein Data Bank under the accession codes 8WUS (termination state), 8WUT (initiation state), 8WUU (pre-initiation state), 8WUV (elongation (16-nt) state), and 8YGJ (elongation (28-nt) state). The cryo-EM density map has been deposited in the Electron Microscopy Data

## Human research participants

Policy information about studies involving human research participants and Sex and Gender in Research.

| | |
|---|---|
| Reporting on sex and gender | n/a |
| Population characteristics | n/a |
| Recruitment | n/a |
| Ethics oversight | n/a |

Note that full information on the approval of the study protocol must also be provided in the manuscript.

# Field-specific reporting

Please select the one below that is the best fit for your research. If you are not sure, read the appropriate sections before making your selection.

☒ Life sciences ☐ Behavioural & social sciences ☐ Ecological, evolutionary & environmental sciences

For a reference copy of the document with all sections, see nature.com/documents/nr-reporting-summary-flat.pdf

# Life sciences study design

All studies must disclose on these points even when the disclosure is negative.

| | |
|---|---|
| Sample size | For cryo-EM analyses, sample sizes were determined by the availability of microscope time and the number of particles on electron microscopy grids enough to obtain a structure at the reported resolution. For cellular experiments, the typical sample sizes were used to ensure reproducibility while maintaining experimental practicality. |
| Data exclusions | For cryo-EM analyses, particles that did not contribute to improving map quality were excluded following the standard classification procedures in cryoSPARC. This is standard practice for structure determination by cryo-EM. For the other experiments, no data was excluded. |
| Replication | For cryo-EM analyses, data processing methods used in cryoSPARC are well-established and reproducible. Biochemical experiments were performed at least three times with successful results. |
| Randomization | For cryo-EM analyses, particles were randomly assigned to half-maps for resolution determination following the standard procedures in cryoSPARC. |
| Blinding | n/a |

# Reporting for specific materials, systems and methods

We require information from authors about some types of materials, experimental systems and methods used in many studies. Here, indicate whether each material, system or method listed is relevant to your study. If you are not sure if a list item applies to your research, read the appropriate section before selecting a response.

## Materials & experimental systems

| n/a | Involved in the study |
|---|---|
| ☒ | ☐ Antibodies |
| ☐ | ☒ Eukaryotic cell lines |
| ☒ | ☐ Palaeontology and archaeology |
| ☒ | ☐ Animals and other organisms |
| ☒ | ☐ Clinical data |
| ☒ | ☐ Dual use research of concern |

## Methods

| n/a | Involved in the study |
|---|---|
| ☒ | ☐ ChIP-seq |
| ☒ | ☐ Flow cytometry |
| ☒ | ☐ MRI-based neuroimaging |

# Eukaryotic cell lines

Policy information about cell lines and Sex and Gender in Research

| | |
|---|---|
| Cell line source(s) | HEK293FT cell line was obtained from ThermoFisher Scientific (R70007). |
| Authentication | The HEK293FT cell line was not authenticated as it was purchased commercially. |
| Mycoplasma contamination | The HEK293FT cell line was not tested for mycoplasma contamination. |
| Commonly misidentified lines<br>(See ICLAC register) | HEK293FT is not in the ICLAC database of misidentified cell lines. |

