## [Peer Review file · Nature]

Manuscript Title: Structural basis for pegRNA-guided reverse transcription by prime editor

Reviewer Comments & Author Rebuttals

Reviewer Reports on the Initial Version:

Referees' comments:

Referee #1:

In this study, Shuto et al. aimed to elucidate the molecular mechanism of pegRNA-guided reverse transcription by the prime editor (PE2). PE2, a tool that combines elements from nSpCas9 and engineered M-MLV RT, could perform precise in vivo genome editing with the help of a pegRNA. Their research addressed the mechanism by studying the SpCas9-M-MLV RT Δ RNaseH-pegRNA-target DNA complex structures. They found that the termination state PE2 can sometimes extend beyond expected sites during reverse transcription, which may cause undesired edits. They also compared different stages of the structures and showed that the M-MLV RT maintains a consistent position relative to SpCas9. Significantly, based on findings, they proposed a strategy to modify pegRNA to better align with the target locus. This work provides significantly important insights into the mechanism of prime editing and sets the stage for developing a more versatile toolbox. The manuscript is logically built, and the figures are appealing and well-structured.

Comments that may help the authors improve their manuscript are provided below.

Major:

1. The length of RTT in the termination state is 6 nt, whereas, for biochemical assay, the length of RTT is 28 nt. Is there a correlation between the efficiency of scaffold-derived incorporation and the length of RTT? Furthermore, in Fig. 3c, all products with added dNTP are 65 nt. The authors should manipulate the length of RTT to observe whether the efficiency of scaffold-derived incorporation changes.
2. For the in vitro biochemical assays, I'm curious whether the efficiency of scaffold-derived incorporation may change by replacing the M-MLV RT with RTs used for PE6 (e.g. PE6a, PE6b).
3. In Fig. 3g, the authors eliminated scaffold-derived incorporations by a modified pegRNA strategy. However, the in vivo prime editing efficacy of PE2 appears low, with only moderate improvement observed in the HEK3 target. The authors may consider validating the modified pegRNA strategy using a more active PE, such as recently published PE6a or PE6b, as discussed. Beyond the 82-84 region, the authors probably can also try to modify the sequences for 79-81, making this region form a perfect base-pairing with the end region of RTT, to further stabilize the gRNA stem and avoid being unwound for reverse transcription. If more convincing data can't be provided in the revision, to avoid misleading the field, I suggest the authors tone down this section (perhaps combine it with the previous section) and move the main figure to the suppl. figure.

4. In Fig. 4i, it is unclear whether the structure of the elongation (28-nt) state was resolved by the authors or if it has been previously reported. No relevant information is listed in Ext. Data Table 1 or citations. Please provide clarification on this issue.

5. Given that the RT Δ RH is positioned consistently relative to SpCas9, especially in the initiation and elongation states, the authors might consider employing 3D Flexible Refinement to enhance the resolution of RT or conducting 3D Variability Analysis to provide additional details on the local variations in the conformation of RT Δ RH compared to SpCas9.

Minor:

1. In order to accurately depict the assembly of different complexes, I recommend that the authors provide images of all protein gels and nucleic acid gels relevant to the reconstitution of each complex.

2. It would be better to indicate the mutation sites of the Cas9 protein in Fig. 1d.

3. There are some mistakes in the annotations within the figures. For instance, a "3'" was mislabeled in Fig. 2b. The authors should carefully check and revise the figures.

4. There are some figure legends in the manuscript that are quite brief and several figures lack essential annotations, such as Fig. 3a and 3d, which are crucial for a comprehensive understanding of the presented data.

5. Page 5: The following text on line 130 (...the 5'side gradually moves closer to Spcas9 as compared to the 3'side.) is ambiguous. Please revise it.

6. On p. 5, lines 120-122, the authors demonstrated a detailed interaction between nucleotides with different residues, while in Fig. 3b, the presented interactions were not as clear as described in the manuscript. A more intuitive and explicit representation of the interactions in Fig. 3b will be highly appreciated.

7. In Fig. 3f, there is a band under the 65-nt products which was not annotated. The authors should explain what this band might be and why this band appeared when using modified pegRNAs.

8. In Ext. Data Fig. 3a, the display of atomic model alignment between SpCas9-pegRNA-target DNA and SpCas9-sgRNA-target DNA is difficult to delineate. A different representation could be more informative.

9. For all biochemical assays, it would be better if the authors could quantify the efficiency of different groups. For instance, in Fig. 4h, the authors should compare the efficiency between WT PE2 and its mutants with efficiency quantification.

10. On line 698, a reference was missing. Please carefully check the manuscript and ensure there was no other such mistakes.

11. The incorporation of RT Δ RH between G1054 and E1055 or T1068 and G1069 in the RuvC domain is intriguing. It would enhance clarity if the authors considered including an additional Ext. Data figure to illustrate the design and AlphaFold-predicted structural models.

Referee #2:

The manuscript, titled "Structural dynamics and engineering of the Prime Editor", investigates the molecular mechanism of prime editing. Prime editors consist of a Cas9 nickase, a reverse transcriptase, and an extended single guide RNA that serves as a template for the synthesis of the edited information. While prime editing is an extremely promising genome editing tool, the lack of structural knowledge about the prime editor complex has been a limitation for developing variants with enhanced efficiency and precision.

The study presents cryo-electron microscopy (cryo-EM) structures of the untethered dCas9 and the M-MLV RT Δ RNaseH in complex with a modified pegRNA in multiple states on a target DNA. This structural analysis revealed how reverse transcription was extended beyond the expected site, causing undesired scaffold insertions at the target loci, as already observed in previous studies. Based on these results, the authors propose a rational pegRNA modification strategy. By aligning the 3'-scaffold sequence with the target locus, this strategy aims to eliminate unintended scaffold insertions while maintaining desired editing efficiency.

Although the study presents valuable insights for the field of genome editing, there are significant shortcomings that need to be resolved before I could recommend publication in Nature.

Major comments:

1) To obtain the structure, several alterations to the commonly used PE2 complex had to be made. PE2 had to be split into untethered Cas9 and the M-MLV RT, the nSpCas9 H840A was altered to nuclease-deficient dSpCas9, and the spacer and RTT had to be modified in order not to comprise complementarity as required for genome editing in the PE2 system. While I understand that these modifications were likely necessary to obtain the structure of a prime editor complex, protein dynamics and structure could vary between tethered and untethered PE, and conclusions drawn from this structure might not apply for prime editing with PE2. This is a downside, as the untethered PE is less efficient than the classical PE2 and rarely applied.

Claims made in the manuscript should, at least, be adapted prior to publication. For example, the authors should clearly state (already in the title and abstract) that they are investigating the untethered PE system. In addition, they should correctly indicate which PE variants were analyzed and also illustrated this in the figures (currently the authors indicated in the figures that nCas9 was used although dCas9 was used to solve the structure).

2) To omit scaffold insertion of three bases, the authors altered the sequence in the stem-loop of the scaffold after the RTT to be homologous to the target strand. Using an in vitro assay the authors

show that this modification abolishes the integration of the scaffold sequence. However, multiple research groups have previously reported scaffold insertions of more than 3 bases at various loci in cell lines (e.g. Anzalone et al., Nature, 2019 (Extended Data 7)). Hence, it seems that their observation of 3 base integrations might be dependent to the sequence of the specific locus. Moreover, the authors propose a very specific solution that might not be translatable to other loci. To make their claims, the authors would need to assess a number of different loci in cell lines, and show that their strategy is broadly applicable also to other sequences.

From the presented data in Fig. 3 it seems that also with non-engineered pegRNAs no scaffold integrations occurred (except for HEK3locus +1 to A, but also there it is unclear if the difference to engineered pegRNAs is significant). Loci in which scaffold integrations occur with non-engineered pegRNAs (see literature) should be chosen, and the difference to engineered pegRNAs should be quantified.

The authors claim that the strategy of engineering the scaffold could be expanded to PE3–PE6 systems. The authors should show this experimentally at several different loci. Additionally, different ratios of pegRNA to PE2 protein could be tested as one would expect that an excess of pegRNA would lead to additional templates at the target sites, where the scaffold is not protected by the dCas9.

3) Fig. 3g: fastq files should be provided. As CRISPResso2 uses filtering in standard settings (mainly to remove unspecific amplicons), it should be further specified how the NGS data was analyzed (CRISPResso settings). In the case of larger scaffold insertions, CRISPResso sometimes mistakenly filters these reads before further analysis. For a more direct/unbiased quantification of scaffold insertions, the authors could also use the script of Anzalone et al.

4) In Ext. Data Fig. 4 g) there is an unusual "bump" in the FSC. Could the FSC be improved by adding more particles or rather "higher quality" particles? Could the authors elaborate on this?

Minor comments:

1) Fig. 1: It would be helpful for the reader not to mix yellow and orange as it is difficult to distinguish these colors.

2) Fig. 4c: It is unclear whether the authors used the dCas9 in combination with or without the RT to resolve the Pre-initiation complex. The order in the figure is misleading, as intuitively the panel would start with Pre-initiation state.

3) The authors state in lines 195–197 that the positions of the PBS-NTS heteroduplex in the preinitiation and initiation states are similar. In the figure it seems that the PBS-NTS heteroduplex in the initiation state is more closely associated with the RuvC domain than in the preinitiation state. Could the authors elaborate on this?

4) Fig. 5: The authors should specify that it is a model of untethered prime editing, hence remove the linker and show what they resolved (dSpCas9, M-MLV RT Δ RNaseH). In addition, the authors did

not show the steric clash leading to dissociation of the M-MLV RT Δ RNaseH, hence this remains speculation which should be stated or removed from the panel.

5) Have the authors also analyzed structures of untethered prime editor with varying length of PBS or RTT? As the authors show in Fig. 4e (elongation state), the RTT-synthesized DNA heteroduplex leads to rearrangements in the PAM-distal target DNA duplex during elongation. It would be interesting to analyze whether there are any structural rearrangements when longer PBS (exceeding the seed region) and RTT are used. Based on their structural data, would the authors assume that pegRNAs with significantly longer RTT and/or PBS result in lower prime editing efficiencies because of steric hindrance with the PAM-distal target DNA duplex? In case the authors don't have data on it they may discuss this point in their manuscript.

Author Rebuttals to Initial Comments:

Responses to reviewers' comments

Reviewer #1:

In this study, Shuto et al. aimed to elucidate the molecular mechanism of pegRNA-guided reverse transcription by the prime editor (PE2). PE2, a tool that combines elements from nSpCas9 and engineered M-MLV RT, could perform precise in vivo genome editing with the help of a pegRNA. Their research addressed the mechanism by studying the SpCas9-M-MLV RTΔRNaseH-pegRNA-target DNA complex structures. They found that the termination state PE2 can sometimes extend beyond expected sites during reverse transcription, which may cause undesired edits. They also compared different stages of the structures and showed that the M-MLV RT maintains a consistent position relative to SpCas9. Significantly, based on findings, they proposed a strategy to modify pegRNA to better align with the target locus. This work provides significantly important insights into the mechanism of prime editing and sets the stage for developing a more versatile toolbox. The manuscript is logically built, and the figures are appealing and well-structured.

Comments that may help the authors improve their manuscript are provided below.

We thank the reviewer for the positive comments. We have addressed the points raised by the reviewer as follows.

Major comments:

1. The length of RTT in the termination state is 6 nt, whereas, for biochemical assay, the length of RTT is 28 nt. Is there a correlation between the efficiency of scaffold-derived incorporation and the length of RTT? Furthermore, in Fig. 3c, all products with added dNTP are 65 nt. The authors should manipulate the length of RTT to observe whether the efficiency of scaffold-derived incorporation changes.

Thank you for the insightful comment. We attempted the structural determination of the termination state using pegRNAs with both 6-nt and 28-nt RTTs, and obtained a high-quality map using the pegRNA with 6-nt RTT. Therefore, we used this map for model building and discussion. In contrast, we used the pegRNA with 28-nt RTT for the biochemical assay to make it easier for readers to discern the differences in reverse transcription product lengths.

Nonetheless, as pointed out by the reviewer, it is crucial to elucidate whether the RTT length affects the efficiency of scaffold-derived incorporations. To investigate the effect of the RTT length, we prepared pegRNAs with 6-, 10-, 20-, and 28-nt RTT sequences. These pegRNAs were designed to contain only “U” at the 5' end, causing reverse transcription to halt at the end of the RTT sequence when using ddATP. Using these pegRNAs, we performed an *in vitro* prime editing assay and compared the lengths of reverse transcription products under ddATP and dNTPs conditions. With all pegRNAs, the reverse transcription products with dNTPs were longer than those with ddATP (Fig. L1). This result indicated that, regardless of the length of the RTT sequence, reverse transcription by the PE2 does not terminate at the RTT terminus, and proceeds to the scaffold region. According to the reviewer’s suggestions, we have added this result to Extended Data Fig. 4a in the revised manuscript.

Fig. L1 | Effects of RTT length on scaffold-derived incorporations.

In vitro prime editing assay using pegRNAs containing 6-, 10-, 20-, and 28-nt RTT sequences. The PE2–pegRNA complexes were added to DNA substrates with dNTPs or with dCTP, dTTP, dGTP, and ddATP (referred to as ddATP for simplicity), and incubated at 37 °C for 10 min. The reaction products were separated on a 10% Novex PAGE TBE–urea gel and then Cy5 fluorescence was visualized. The experiments were repeated three times with similar results.

2. For the *in vitro* biochemical assays, I’m curious whether the efficiency of scaffold-derived incorporation may change by replacing the M-MLV RT with RTs used for PE6 (e.g. PE6a, PE6b).

Thank you for the comment. We expressed the recently reported four PE6 constructs with different RTs (PE6a–d) and successfully purified them¹ (Fig. L2a). To assess whether other PEs induce scaffold-derived incorporations, we performed an *in vitro* prime editing assay using

these purified PEs. Our results showed that PE6a and PE6d efficiently generated DNA products within 10 min, whereas PE6b and PE6c did not complete reverse transcription within the same timeframe, suggesting that PE6b and PE6c exhibit lower reverse transcription efficiencies compared to other PEs, at least under our experimental conditions (Fig. L2b). However, we confirmed that PE6a–d also generate the same length reverse transcription products as PE2, under 60-min conditions (Fig. L2c). These findings imply that other RTs, like the M-MLV RT, invade the scaffold region without terminating at the end of the RTT sequence. According to the reviewer’s comments, we have added these results in the revised manuscript (Extended Data Fig. 10f).

Fig. L2 | *In vitro* prime editing analysis of PE6a–d proteins.

(a) SDS-PAGE gel of purified PE2 and PE6a–d proteins.

(b and c) *In vitro* prime editing assay using PE2 and PE6a–d proteins and the pegRNA containing the 28-nt RTT sequence. The PE2/PE6a–d–pegRNA complex was added to DNA substrates with dNTPs and incubated at 37 °C for 10 min (b) and 60 min (c). The 62-nt marker represents the product transcribed up to the end of the RTT, while the 65-nt marker represents the product transcribed up to U94 in the pegRNA scaffold.

3. In Fig. 3g, the authors eliminated scaffold-derived incorporations by a modified pegRNA strategy. However, the *in vivo* prime editing efficacy of PE2 appears low, with only moderate

improvement observed in the HEK3 target. The authors may consider validating the modified pegRNA strategy using a more active PE, such as recently published PE6a or PE6b, as discussed. Beyond the 82-84 region, the authors probably can also try to modify the sequences for 79-81, making this region form a perfect base-pairing with the end region of RTT, to further stabilize the gRNA stem and avoid being unwound for reverse transcription. If more convincing data can't be provided in the revision, to avoid misleading the field, I suggest the authors tone down this section (perhaps combine it with the previous section) and move the main figure to the suppl. figure.

According to your helpful comments, we performed a prime editing assay using a more active PE3 system under the four previously reported conditions in HEK293 cells². As in the PE2 system, the modified pegRNAs exhibited comparable desired editing efficiency to the wild-type pegRNA in the PE3 system (Fig. L3). However, the modified pegRNAs also induced long undesired insertions, such as those derived from the RTT sequence or scaffold sequences longer than three nucleotides, at levels comparable to the wild-type pegRNA (Fig. L3). These findings suggest that while our modified pegRNA can eliminate three-nucleotide scaffold incorporations, it cannot exclude other common undesired incorporations. Therefore, our modified pegRNA is only effective at target sites where short scaffold-derived insertions are predominant. Accordingly, as you suggested, we have toned down this section and moved the main figure to the Extended figures (Extended Data Fig. 4c–e).

Fig. L3 | *In vivo* prime editing analysis of PE2 and PE3 systems.

The desired edit and the undesired incorporation efficiencies of the PE2 system with five conditions (left) and the PE3 system with four conditions (right) using the wild-type pegRNA and modified pegRNA in HEK293FT cells. Data are mean \pm s.d. (n = 3 biologically independent samples).

4. In Fig. 4i, it is unclear whether the structure of the elongation (28-nt) state was resolved by the authors or if it has been previously reported. No relevant information is listed in Ext. Data Table 1 or citations. Please provide clarification on this issue.

Thank you for the comments. The structure of the elongation (28-nt) state was also determined in this study. According to the reviewer's suggestion, we have added the relevant information to Extended Data Fig. 9 and Extended Data Table 1 in the revised manuscript.

5. Given that the RT Δ RH is positioned consistently relative to SpCas9, especially in the initiation and elongation states, the authors might consider employing 3D Flexible Refinement to enhance the resolution of RT or conducting 3D Variability Analysis to provide additional details on the local variations in the conformation of RT Δ RH compared to SpCas9.

According to the reviewer's helpful suggestion, we performed 3D Flexible Refinement³ (3DFlex) using the final particle sets of the initiation states and obtained a new density map (3DFlex map). Although this map improved the local resolution around the RT in the overall map, it was not superior to the original RT-focused map obtained with local refinement (composite map) (Fig. L4). Likewise, the 3D Variability Analysis⁴ provided only minor local variations, which do not offer informative structural insights. Therefore, we opted not to include this result in the revised manuscript.

Fig. L4 | Comparison of the cryo-EM density maps in the initiation state.

Overall (left), 3DFlex (center), and composite (right) maps in the initiation state. The density corresponding to RT Δ RH is enclosed within a black circle on each map.

Minor comments:

1. In order to accurately depict the assembly of different complexes, I recommend that the authors provide images of all protein gels and nucleic acid gels relevant to the reconstitution of each complex.

Thank you for the helpful comment. We have included size-exclusion chromatography profiles and SDS-PAGE/urea-PAGE gels relevant to the reconstitution of each complex in the revised manuscript.

2. It would be better to indicate the mutation sites of the Cas9 protein in Fig. 1d.

According to the reviewer's suggestion, we added the mutation sites of SpCas9 (D10A/H840A) and M-MLV RT (D200N/T306K/W313F/T330P) in Fig. 1d of the revised manuscript.

3. There are some mistakes in the annotations within the figures. For instance, a " 3' " was mislabeled in Fig. 2b. The authors should carefully check and revise the figures.

We fixed them in the revised manuscript and carefully checked and revised the other parts.

4. There are some figure legends in the manuscript that are quite brief and several figures lack essential annotations, such as Fig. 3a and 3d, which are crucial for a comprehensive understanding of the presented data.

Thank you for the comments. We have added detailed information in Fig. 3a, 3b, and 3d.

5. Page 5: The following text on line 130 (...the 5' side gradually moves closer to Spcas9 as compared to the 3' side.) is ambiguous. Please revise it.

As suggested, in the revised manuscript we have changed the statement “Our structure shows that the 3' end of the scaffold region (nucleotides U94–C96) base pairs with nucleotides A84–G82 to form a stem loop structure, and the 5' side gradually moves closer to SpCas9 as compared to the 3' side” to “Our structure shows that U94 of the pegRNA is positioned close to SpCas9, suggesting that M-MLV RT may not proceed further.”

6. On p. 5, lines 120-122, the authors demonstrated a detailed interaction between nucleotides with different residues, while in Fig. 3b, the presented interactions were not as clear as described in the manuscript. A more intuitive and explicit representation of the interactions in Fig. 3b will be highly appreciated.

Thank you for the valuable comment. To illustrate the interaction between M-MLV RT and the scaffold region more clearly, we have adjusted the angle of Fig. 3b and provided detailed explanations in that figure legend.

7. In Fig. 3f, there is a band under the 65-nt products which was not annotated. The authors should explain what this band might be and why this band appeared when using modified pegRNAs.

Thank you for the comments. The bands observed under the 65-nt products with modified pegRNAs in Fig. 1f in the original manuscript were likely DNA-templated products, as observed in Fig. 1c. To confirm this, we repeated the *in vitro* prime editing assay and observed clear bands in the absence of pegRNA at positions similar to those in the modified pegRNAs of the original gels (Fig. L5). This result suggests that the bands under the 65-nt products represent DNA-templated products. However, despite repeating the experiment three times, we did not observe DNA-templated products using three modified pegRNAs (Fig. L5). We do not have a clear answer as to why DNA-template products appeared in previous experiments. However, this result indicates that the modified pegRNA induces reverse transcription with efficiency comparable to that of the wild-type pegRNA, which is consistent with our *in vivo* analysis. Therefore, we have included the results of this experiment in the revised manuscript (Extended Data Fig. 4d).

Fig. L5 | *In vitro* prime editing assay using the wild-type and three modified pegRNAs.

The PE2 or PE2–pegRNA complexes were added to DNA substrates with dNTPs and incubated at 37 °C for 10 min. In the absence of the pegRNA (DNA only), PE2 generates DNA-templated products (51-nt). The comparison between the original and revised results suggests that the bands observed under the 65-nt products with modified pegRNAs in the original manuscript are likely DNA-templated products. However, we did not observe such bands when using modified pegRNAs in the revised experiment, for an unknown reason.

8. *In Ext. Data Fig. 3a, the display of atomic model alignment between SpCas9-pegRNA-target DNA and SpCas9-sgRNA-target DNA is difficult to delineate. A different representation could be more informative.*

According to the reviewer’s helpful comments, we changed the ribbon representation in Extended Data Fig. 3a into a cylinder representation for clarity in the revised manuscript.

9. *For all biochemical assays, it would be better if the authors could quantify the efficiency of different groups. For instance, in Fig. 4h, the authors should compare the efficiency between WT PE2 and its mutants with efficiency quantification.*

We appreciate the reviewer’s helpful suggestion. The reverse transcription efficiency of each group was calculated using the equation: $a/(a+b+c) * 100$, where “a” is the RTT-templated reverse transcription product band intensity, “b” is the substrate band intensity, and “c” is the

DNA-templated band intensity. The band intensity was quantified using Image J⁵. We show the efficiency in Figs. 1c, 4h and Extended Data Fig. 4d in the revised manuscript.

10. On line 698, a reference was missing. Please carefully check the manuscript and ensure there was no other such mistakes.

We fixed them in the revised manuscript and carefully checked the other parts.

11. The incorporation of RTΔRH between G1054 and E1055 or T1068 and G1069 in the RuvC domain is intriguing. It would enhance clarity if the authors considered including an additional Ext. Data figure to illustrate the design and AlphaFold-predicted structural models.

Thank you for the insightful comments. We predicted the two mutant structures, G1054–RTΔRH–E1055 and T1068–RTΔRH–G1069, using AlphaFold2⁶. In these structures, the RTΔRH in both models is anchored around the position where the PBS–NTS heteroduplex forms, as intended (Fig. L6). We have added the AlphaFold-predicted structures and relevant information in Extended Data Fig. 8h in the revised manuscript.

Fig. L6 | AlphaFold2-prediction models of PE2 variants.

The AlphaFold-prediction models of G1054–RTΔRH–E1055 (left) and T1068–RTΔRH–G1069 (right) PE2 variants. The nucleic acid of the initiation state was mapped onto these AlphaFold models. G1054, E1055, T1068, and G1069 are shown as orange space-filling models. The five amino-acid linkers are colored black.

Reviewer #2:

The manuscript, titled "Structural dynamics and engineering of the Prime Editor", investigates the molecular mechanism of prime editing. Prime editors consist of a Cas9 nickase, a reverse transcriptase, and an extended single guide RNA that serves as a template for the synthesis of the edited information. While prime editing is an extremely promising genome editing tool, the lack of structural knowledge about the prime editor complex has been a limitation for developing variants with enhanced efficiency and precision.

The study presents cryo-electron microscopy (cryo-EM) structures of the untethered dCas9 and the M-MLV RTARNaseH in complex with a modified pegRNA in multiple states on a target DNA. This structural analysis revealed how reverse transcription was extended beyond the expected site, causing undesired scaffold insertions at the target loci, as already observed in previous studies. Based on these results, the authors propose a rational pegRNA modification strategy. By aligning the 3'-scaffold sequence with the target locus, this strategy aims to eliminate unintended scaffold insertions while maintaining desired editing efficiency.

Although the study presents valuable insights for the field of genome editing, there are significant shortcomings that need to be resolved before I could recommend publication in Nature.

We thank the reviewer for the comments. We have addressed the points raised by the reviewer as follows.

Major comments:

1. To obtain the structure, several alterations to the commonly used PE2 complex had to be made. PE2 had to be split into untethered Cas9 and the M-MLV RT, the nSpCas9 H840A was altered to nuclease-deficient dSpCas9, and the spacer and RTT had to be modified in order not to comprise complementarity as required for genome editing in the PE2 system. While I understand that these modifications were likely necessary to obtain the structure of a prime editor complex, protein dynamics and structure could vary between tethered and untethered PE, and conclusions drawn from this structure might not apply for prime editing with PE2. This is a downside, as the untethered PE is less efficient than the classical PE2 and rarely applied.

Claims made in the manuscript should, at least, be adapted prior to publication. For example, the authors should clearly state (already in the title and abstract) that they are investigating the untethered PE system. In addition, they should correctly indicate which PE variants were analyzed and also illustrated this in the figures (currently the authors indicated in the figures that nCas9 was used although dCas9 was used to solve the structure).

We thank the reviewer for the insightful comments. We introduced several modifications to the PE2 system to determine the structure of the prime editor. Specifically, we utilized an untethered PE system, where SpCas9 and M-MLV RT are not tethered by a linker. While the reviewer argued that this untethered PE system is less efficient, previous studies by Liu *et al.* and Grünewald *et al.* have reported that this system shows comparable activity to the PE2 system^{7,8}. Considering its recent development, the limited application of the untethered PE system is understandable. Furthermore, in the PE2 system, SpCas9 and M-MLV RT are connected by a 33-amino acid linker. Based on the AlphaFold2 prediction-model of PE2 shown in Fig. L7, this linker has sufficient length to be an intrinsically disordered structure and would not restrict the structural dynamics of SpCas9 and M-MLV RT. Therefore, we believe that the prime editing mechanism proposed from our structural and biochemical analyses in this study is applicable not only to the untethered PE system but also to the PE2 system. Indeed, our biochemical analysis revealed that PE2, like the untethered PE, induces three-nucleotide incorporations of the scaffold sequences. However, as the reviewer pointed out, there were misleading aspects regarding the constructs used for structure determination in the original manuscript. Therefore, we have explicitly revised the constructs used for structural analysis in the revised figure and clarified in the abstract and main text that our structural analysis was conducted using the untethered PE system.

Fig. L7 | AlphaFold2-prediction models of PE2.

The top two highly accurate models predicted for the PE2 structural model by AlphaFold2⁶. Model 1 shows M-MLV RT located on the REC lobe side of SpCas9, while Model 2 shows it on the NUC lobe side. The linker region adopts a loose conformation in both models. These observations suggest that in the PE2 system, the linker does not affect the dynamics of SpCas9 and M-MLV RT.

2. To omit scaffold insertion of three bases, the authors altered the sequence in the stem-loop of the scaffold after the RTT to be homologous to the target strand. Using an *in vitro* assay the authors show that this modification abolishes the integration of the scaffold sequence. However, multiple research groups have previously reported scaffold insertions of more than 3 bases at various loci in cell lines (e.g. Anzalone *et al.*, *Nature*, 2019 (Extended Data 7)). Hence, it seems that their observation of 3 base integrations might be dependent to the sequence of the specific locus. Moreover, the authors propose a very specific solution that might not be translatable to

other loci. To make their claims, the authors would need to assess a number of different loci in cell lines, and show that their strategy is broadly applicable also to other sequences.

From the presented data in Fig. 3 it seems that also with non-engineered pegRNAs no scaffold integrations occurred (except for HEK3locus +1 to A, but also there it is unclear if the difference to engineered pegRNAs is significant). Loci in which scaffold integrations occur with non-engineered pegRNAs (see literature) should be chosen, and the difference to engineered pegRNAs should be quantified.

The authors claim that the strategy of engineering the scaffold could be expanded to PE3–PE6 systems. The loci. Additionally, different ratios of pegRNA to PE2 protein could be tested as one would expect that an excess of pegRNA would lead to additional templates at the target sites, where the scaffold is not protected by the dCas9.

Thank you for your valuable comments. According to your next comment (Major #3), we re-analyzed the PE2 data with the script utilized by Anzalone *et al*². We found that both wild-type and modified pegRNAs induce long undesired incorporations, including those derived from the RTT sequence or scaffold sequences longer than three nucleotides (Fig. L8a). We also performed a prime editing assay using a more active PE3 system under the four previously reported conditions in HEK293 cells². In the PE2 system, while the modified pegRNAs exhibited comparable desired editing efficiencies to the wild-type (WT) pegRNA in the PE3 system, they also induced long undesired insertions at levels similar to the WT pegRNA (Fig. L8a). These findings suggest that our modified pegRNA can eliminate three-nucleotide scaffold incorporations, but cannot exclude other prevalent undesired incorporations. Therefore, our modified pegRNA is only effective at target sites where short scaffold-derived insertions are predominant. Accordingly, we have toned down this section and moved the main figure to the Extended figures (Extended Data Fig. 4c–e).

Furthermore, to confirm whether an excess of pegRNA would result in additional templates at the target sites, we performed the prime editing analysis using different amounts of transfected pegRNA. We observed a significant decrease in the prime editing efficiency when transfecting reduced amounts of pegRNA, while the undesired incorporations remained relatively unchanged (Fig. L8b). This finding suggests that undesired incorporations are not solely dependent on the free-pegRNA that does not form a complex with nSpCas9. Elucidating the mechanisms inducing these insertions and developing strategies to eliminate them will be

crucial for future applications of prime editing. We have mentioned this in the Discussion section of the revised manuscript.

Fig. L8 | *In vivo* prime editing analysis of PE2 and PE3 systems.

(a) The desired edits and the undesired incorporation efficiencies of the PE2 system with five conditions (left) and the PE3 system with four conditions (right) using the wild-type and modified pegRNAs in HEK293FT cells. Data are mean \pm s.d. ($n = 3$ biologically independent samples).

(b) Comparison of desired edits and undesired incorporations in the PE3 system with varying amounts of pegRNA. A total of 135 ng of PE3 plasmid was consistently transfected, while the pegRNA plasmid was transfected at 50, 25, and 12.5 ng into each well. Data are mean \pm s.d.

3. *Fig. 3g: fastq files should be provided. As CRISPResso2 uses filtering in standard settings (mainly to remove unspecific amplicons), it should be further specified how the NGS data was analyzed (CRISPResso settings). In the case of larger scaffold insertions, CRISPResso sometimes mistakenly filters these reads before further analysis. For a more direct/unbiased quantification of scaffold insertions, the authors could also use the script of Anzalone et al.*

Thank you for the comments. We have uploaded all the fastq files in the NCBI under the accession code PRJNA1084104. For CRISPResso2⁹, we basically use the default settings. We use 60% as the Minimum homology for alignment to an amplicon. The pegRNA extension quantification window size is 5 bp. We do not perform any quality filtering and trimming. For the settings of exclude bp from the left and right side of the amplicon sequence for the quantification of the mutations, we use 15 bp for both by default. We appreciate your suggestion to use the script of Anzalone *et al.* to analyze the scaffold insertions². In the revised manuscript, all of the scaffold insertion results are now generated by this script.

4. *In Ext. Data Fig. 4 g) there is an unusual "bump" in the FSC. Could the FSC be improved by adding more particles or rather "higher quality" particles? Could the authors elaborate on this?*

In general, the "bump" in the FSC is relatively common when mask-focused regions exhibit flexibility or when the mask is too tight. Regarding the former point, since the RT interacts only with the PBS–NTS heteroduplex flanked by single-strand nucleotides, the focused RT region indeed exhibits flexibility in the initiation complex. Therefore, it would not be unusual to observe the "bump". In this case, as the reviewer mentioned, the map and FSC might be improved by using more particles. However, the picked particles (118,125 particles) were abundant in this analysis, and the current map quality is sufficient for our results and discussion. Regarding the latter point, the mask used for the FSC calculation in the original manuscript was a little too tight. Therefore, we re-calculated the FSC and resolution for the initiation complex using a looser mask, and obtained a new FSC plot, which still includes a "bump" due to its

flexibility (Fig. L9). According to the reviewer's suggestion, we have replaced the FSC in Extended Data Fig. 4g with a new one in the revised manuscript.

Fig. L9 | Fourier shell correlation (FSC) curves.

FSC curves for the local 3D reconstruction of the initiation state, calculated using the original mask (left) and the revised mask (right). The gold standard cutoff (FSC = 0.143) is marked with the black dotted line.

Minor comments:

1. Fig. 1: It would be helpful for the reader not to mix yellow and orange as it is difficult to distinguish these colors.

We thank the reviewer for the helpful comment. We now use pink for PBS and yellow for RTT throughout the revised manuscript so readers can easily distinguish these regions.

2. Fig. 4c: It is unclear whether the authors used the dCas9 in combination with or without the RT to resolve the Pre-initiation complex. The order in the figure is misleading, as intuitively the panel would start with Pre-initiation state.

Thank you for the comments. We determined the pre-initiation complex without the RT. To clarify this, we have explicitly indicated the phrase “without RT” within Fig. 4c. Although we understand the reviewer's opinion that the order in Fig. 4 is confusing, it is difficult to rearrange the order in this figure due to the logical flow of the manuscript. Therefore, we did not change the order of the panels in Fig. 4.

3. The authors state in lines 195–197 that the positions of the PBS-NTS heteroduplex in the preinitiation and initiation states are similar. In the figure it seems that the PBS-NTS heteroduplex in the initiation state is more closely associated with the RuvC domain than in the preinitiation state. Could the authors elaborate on this??

We thank the reviewer for the insightful comment. As the reviewer points out, the PBS–NTS heteroduplex in the initiation state is slightly closer to the RuvC domain than that in the pre-initiation state. However, the heteroduplex in the initiation state is still approximately 10 Å away from the RuvC domain and does not directly interact with it. Considering the overall structure of the prime editor, we believe that our statement that the position of the heteroduplex is similar in these two states is accurate.

4. Fig. 5: The authors should specify that it is a model of untethered prime editing, hence remove the linker and show what they resolved (dSpCas9, M-MLV RTΔRNaseH). In addition, the authors did not show the steric clash leading to dissociation of the M-MLV RTΔRNaseH, hence this remains speculation which should be stated or removed from the panel.

As we explained in our response to the reviewer's major comment #1, we believe that the model of prime editing proposed from our structural analysis is common to both the untethered PE and the PE2 system. Therefore, it would be acceptable to represent the model using nSpCas9, M-MLV RT, and the linker. As the reviewer pointed out, our model that the steric clash between SpCas9 and M-MLV RT leads to the termination of reverse transcription and the dissociation of the M-MLV RT from the pegRNA is purely speculation. Accordingly, we have explicitly stated this in both the main text and figure legend in the revised manuscript.

5. Have the authors also analyzed structures of untethered prime editor with varying length of PBS or RTT? As the authors show in Fig. 4e (elongation state), the RTT-synthesized DNA heteroduplex leads to rearrangements in the PAM-distal target DNA duplex during elongation. It would be interesting to analyze whether there are any structural rearrangements when longer PBS (exceeding the seed region) and RTT are used. Based on their structural data, would the authors assume that pegRNAs with significantly longer RTT and/or PBS result in lower prime editing efficiencies because of steric hindrance with the PAM-distal target DNA duplex? In case the authors don't have data on it they may discuss this point in their manuscript

Thank you for the valuable comments. PegRNAs with significantly longer RTTs exhibit lower prime editing efficiency, which is a disadvantage in the application of prime editors. Interestingly, the elongation state with the 28-nt RTT revealed that the rearrangement of the PAM-distal target DNA duplex induces the disruption of the base pairs at the end of the guide RNA–target DNA heteroduplex (Fig. L10). These observations suggest that further reverse transcription by the RTT leads to additional dissociation between the guide RNA and target DNA heteroduplex, resulting in the detachment of the prime editor from the target site. This may be one of the reasons why the prime editor exhibits low efficiency in inserting long sequences. We have added these statements in the Discussion section of the revised manuscript.

Fig. L10 | Structure of the 28-nt Elongation state.

(a) Overall structure of the SpCas9–RTΔRH–pegRNA–target DNA complex in the elongation (28-nt) state. The disordered regions are indicated as dotted lines.

(b) Close-up views of the PBS–NTS heteroduplex, the PAM-distal duplex, and the end of the guide RNA–target DNA heteroduplex. As the reverse transcription of the RTT sequence progresses, the PBS–NTS heteroduplex is pushed in the opposite direction from the RTΔRH (black arrow), resulting in the rearrangement of the PAM-distal duplex. This rearrangement induces the disruption of the base pairs at the end of the guide RNA–target DNA heteroduplex (G1–dC20 and G2–dC19).

References

1. Doman, J. L. *et al.* Phage-assisted evolution and protein engineering yield compact, efficient prime editors. *Cell* **186**, 3983–4002.e26 (2023).
2. Anzalone, A. V. *et al.* Search-and-replace genome editing without double-strand breaks or donor DNA. *Nature* **576**, 149–157 (2019).

3. Punjani, A. & Fleet, D. J. 3DFlex: determining structure and motion of flexible proteins from cryo-EM. *Nat. Methods* (2023) doi:10.1038/s41592-023-01853-8.
4. Punjani, A. & Fleet, D. J. 3D variability analysis: Resolving continuous flexibility and discrete heterogeneity from single particle cryo-EM. *J. Struct. Biol.* **213**, 107702 (2021).
5. Schneider, C. A., Rasband, W. S. & Eliceiri, K. W. NIH Image to ImageJ: 25 years of image analysis. *Nat. Methods* **9**, 671–675 (2012).
6. Jumper, J. *et al.* Highly accurate protein structure prediction with AlphaFold. *Nature* **596**, 583–589 (2021).
7. Liu, B. *et al.* A split prime editor with untethered reverse transcriptase and circular RNA template. *Nat. Biotechnol.* **40**, 1388–1393 (2022).
8. Grünewald, J. *et al.* Engineered CRISPR prime editors with compact, untethered reverse transcriptases. *Nat. Biotechnol.* **41**, 337–343 (2022).
9. Clement, K. *et al.* CRISPResso2 provides accurate and rapid genome editing sequence analysis. *Nat. Biotechnol.* **37**, 224–226 (2019).

Reviewer Reports on the First Revision:

Referees' comments:

Referee #1:

The authors have performed several additional experiments and provided deeper analyses of the data presented. The majority of my concerns have been addressed. Compared to the initial version, they have improved the quality and presentation of their results and added valuable experiments to further substantiate their conclusions. The added experiments on exploring the correlation between RTT length and scaffold-derived incorporation efficiency, examining the relationship between different types of RTs and incorporation efficiency, assessing the impact of pegRNA quantity on prime editing efficiency, and expanding on structural predictions have significantly enhanced the manuscript.

I recommend this article for publication in Nature, although there are a few more comments that require clarification.

1. In Ext. Data Fig. 4e and Fig. L3, the authors demonstrated that the modified pegRNA could only eliminate three-nucleotide scaffold incorporations at target sites where short scaffold-derived insertions are predominant. However, the *in vivo* prime analysis appears insufficient to fully support this conclusion. I suggest that the authors categorize the 'Undesired incorporations' into 'three-nucleotide scaffold incorporations (3-nt)' and 'scaffold incorporations longer than three nucleotides (>3-nt)', and annotate which sites belong to 'target sites where short scaffold-derived insertions are predominant'.

2. In the Response to Minor comments. 7, 'Fig. 1f' should be corrected to 'Fig. 3f'. The authors should carefully check and revise the manuscript before publication.

Referee #2:

The authors have satisfactorily addressed most of my initial concerns.

I would strongly urge them though to explicitly mention in the title and abstract that their study resolved the structure of the untethered Prime Editor system (the abstract should also specify that they used dCas9 instead nCas9). Despite the fact that the untethered prime editing system (using nSpCas9 and RT-MLV) works well in HEK cells transfected with plasmids, it has been reported at conferences that the editing efficiency with untethered prime editors is markedly lower compared to the traditional prime editor system when delivered as mRNA. To prevent discouragement of future research, it is crucial to clearly communicate that the structure of the tethered prime editing system has not yet been resolved.

Following our earlier feedback, the authors now used a more unbiased quantification of scaffold insertions. This analysis indicates that altering the scaffold does not mitigate the occurrence of longer scaffold insertions. As such, this part of the manuscript becomes less impactful, and the structural analysis becomes the focal point of this study, which I still recommend for publication at Nature.

A minor suggestion for Ext. Data Fig. 4e: please clarify the color coding in the plots to avoid the misconception that the left panel represents “Wild-type pegRNA” exclusively, and the right panel “Modified pegRNA”.

Author Rebuttals to First Revision:

Responses to reviewers' comments

Referee #1: The authors have performed several additional experiments and provided deeper analyses of the data presented. The majority of my concerns have been addressed. Compared to the initial version, they have improved the quality and presentation of their results and added valuable experiments to further substantiate their conclusions. The added experiments on exploring the correlation between RTT length and scaffold-derived incorporation efficiency, examining the relationship between different types of RTs and incorporation efficiency, assessing the impact of pegRNA quantity on prime editing efficiency, and expanding on structural predictions have significantly enhanced the manuscript.

I recommend this article for publication in Nature, although there are a few more comments that require clarification.

We thank the reviewer for the positive comments.

1. In Ext. Data Fig. 4e and Fig. L3, the authors demonstrated that the modified pegRNA could only eliminate three-nucleotide scaffold incorporations at target sites where short scaffold-derived insertions are predominant. However, the in vivo prime analysis appears insufficient to fully support this conclusion. I suggest that the authors categorize the 'Undesired incorporations' into 'three-nucleotide scaffold incorporations (3-nt)' and 'scaffold incorporations longer than three nucleotides (>3-nt)', and annotate which sites belong to 'target sites where short scaffold-derived insertions are predominant'.

Thank you for the insightful comments. As the reviewer's pointed out, to emphasize our conclusion, we should categorize the undesired incorporations and annotate the target site in which the short scaffold incorporations are predominant. However, it is difficult to objectively distinguish the short scaffold-derived incorporations from other short insertions, such as those originating from RTT duplicates or mixtures of random indels, due to the short length of the modified sequence. Therefore, we maintain the previous plots with the total undesired incorporations and have changed the statement "These results indicate that this strategy is only effective at target sites where short scaffold-derived incorporations are prevalent" to "These results suggest that these types of incorporations are dominant in the target sites, and this

strategy would be only effective at target sites where the short scaffold-derived incorporations are prevalent” for clarity in the revised manuscript.

2. In the Response to Minor comments. 7, ‘Fig. 1f’ should be corrected to ‘Fig. 3f’. The authors should carefully check and revise the manuscript before publication.

Thank you for the comment. We carefully checked the revised manuscript.

Referee #2: The authors have satisfactorily addressed most of my initial concerns.

I would strongly urge them though to explicitly mention in the title and abstract that their study resolved the structure of the untethered Prime Editor system (the abstract should also specify that they used dCas9 instead nCas9). Despite the fact that the untethered prime editing system (using nSpCas9 and RT-MLV) works well in HEK cells transfected with plasmids, it has been reported at conferences that the editing efficiency with untethered prime editors is markedly lower compared to the traditional prime editor system when delivered as mRNA. To prevent discouragement of future research, it is crucial to clearly communicate that the structure of the tethered prime editing system has not yet been resolved.

As we stated in the previous responses, the linker connected SpCas9 and M-MLV RT has sufficient length to be an intrinsically disordered structure and would not restrict the structural dynamics of SpCas9 and M-MLV RT. Therefore, we believe that the prime editing mechanism proposed from our structural and biochemical analyses in this study is applicable not only to the untethered PE system but also to the tethered PE2 system. Indeed, Liu *et al.*, has reported that the untethered system shows comparable activity to the tethered system when delivered as mRNAs and RNPs in human cells¹. Although the reviewer claimed that there is a difference in activity between tethered and untethered PE system when delivered as mRNA (we have not grasped this data), this is likely due to differences in cellular localization and expression levels rather than structural dynamics. Therefore, we maintain the expressions in the revised manuscript.

Following our earlier feedback, the authors now used a more unbiased quantification of scaffold insertions. This analysis indicates that altering the scaffold does not mitigate the occurrence of longer scaffold insertions. As such, this part of the manuscript becomes less

impactful, and the structural analysis becomes the focal point of this study, which I still recommend for publication at Nature.

We thank the reviewer for the positive comment.

A minor suggestion for Ext. Data Fig. 4e: please clarify the color coding in the plots to avoid the misconception that the left panel represents “Wild-type pegRNA” exclusively, and the right panel “Modified pegRNA”.

We thank the reviewer for the helpful comment. To avoid the misconception, we have rearranged the placement of the legend in the revised manuscript.

Thank you again for your thoughtful consideration, and we hope that the revised manuscript is now acceptable for publication in *Nature*.

Sincerely,
Osamu

References

1. Liu, B. *et al.* A split prime editor with untethered reverse transcriptase and circular RNA template. *Nat. Biotechnol.* **40**, 1388–1393 (2022).